# Beyond universality in repulsive SU(N) Fermi gases

**Jordi Pera, Joaquim Casulleras and Jordi Boronat**

Departament de Física, Campus Nord B4-B5, Universitat Politècnica de Catalunya,
E-08034 Barcelona, Spain

## Abstract

Itinerant ferromagnetism in dilute Fermi gases is predicted to emerge at values of the gas parameter where second-order perturbation theory is not accurate enough to properly describe the system. We have revisited perturbation theory for SU(N) fermions and derived its generalization up to third order both in terms of the gas parameter and the polarization. Our results agree satisfactorily with quantum Monte Carlo results for hard-sphere and soft-sphere potentials for $S = 1/2$. Although the nature of the phase transition depends on the interaction potential, we find that for a hard-sphere potential a phase transition is guaranteed to occur. While for $S = 1/2$ we observe a quasi-continuous transition, for spins 3/2 and 5/2, a first-order phase transition is found. For larger spins, a double transition (combination of continuous and discontinuous) occurs. The critical density reduces drastically when the spin increases, making the phase transition more accessible to experiments with ultracold dilute Fermi gases. Estimations for Fermi gases of Yb and Sr with spin 5/2 and 9/2, respectively, are reported.

| | |
|---|---|
| Received | 01-06-2023 |
| Accepted | 27-06-2024 |
| Published | 01-08-2024 |

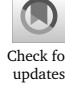

# 1   Introduction

The well-known Stoner model of itinerant ferromagnetism [1] predicts a transition to a ferromagnetic phase for an electron gas when density is increased, due to the interplay between potential and kinetic energies. Trapped cold Fermi gases offer an ideal platform to study itinerant ferromagnetism that, in real materials, has proven to be extremely elusive [2, 3]. This transition must happen in the repulsive branch, which is metastable with respect to the formation of spin-up spin-down dimers [4]. A first pioneering observation of the ferromagnetic transition in cold Fermi gases [5] was revised and concluded that pair formation hinders the achievement of the gas parameters required for observing this transition [6]. Recently, it has been reported that the ferromagnetic state is effectively observed in a Fermi $^7$Li gas around a gas parameter $x = k_F a_0 \simeq 1$, with $k_F$ the Fermi momentum and $a_0$ the $s$-wave scattering length [7]. This transition has been extensively studied from a theoretical point of view, mainly using quantum Monte Carlo methods [8–15]. These numerical estimations agree to localize the ferromagnetic transition around $x \simeq 1$.

Dilute Fermi gases can also be studied using perturbation theory. At first order, the famous Stoner model [1] predicts a continuous phase transition for $S = 1/2$ and a first-order one for $S > 1/2$. As the Stoner model (Hartree-Fock approximation) in not accurate enough when the gas parameter grows, a second-order theory was developed long time ago [16–19], but not applicable when the number of particles in each spin is different. Recently, this perturbative correction for SU(N) gases has been derived in a fully analytical form in terms of both polarization and gas parameter [20]. Previous results by Kanno [21] were limited to the hard-sphere potential and $S = 1/2$. At second order one observes that the ferromagnetic transition for $S = 1/2$ turns to be discontinuous, breaking the asymmetry produced by the Stoner model for different spin values.

Second-order perturbation energies improve substantially the Stoner model but still are not accurate enough to study Fermi gases close to the expected critical densities. Old results, at third-order of perturbation theory, are reported in Ref. [22]. However, to fully characterize the magnetic behavior of the Fermi gases it is fundamental to know the dependency of the energy on the gas parameter but also on the spin polarization [23, 24]. In the present work, we derive the energy as a function of both parameters. Unlike the second order, we have not managed to derive a fully analytical expression. It is worth noticing that, going beyond second order breaks universality, meaning that the energy dependence is no longer solely determined by the $s$-wave scattering length. As we will demonstrate, two additional scattering parameters come into play in the description: the $s$-wave effective range and the $p$-wave scattering length. It is worth mentioning that, for a spin balanced gas, the fourth order term has been fully derived in Ref. [25].

In recent years, the experimental production of SU(N) fermions has renewed the theoretical interest in their study. In particular, Ytterbium [26], with spin 5/2, and Strontium [27],

with spin 9/2, are now available for studying Fermi gases with spin degeneracy never achieved before. Importantly, Cazalilla *et al.* [28, 29] showed that Fermi gases made of alkaline atoms with two electrons in the external shell, such as $^{173}$Yb, present an SU(N) emergent symmetry. They also argued that the ferromagnetic transition must be of first order when $S > 1/2$, based on the significant dissimilarities in the mathematical structure between SU(N>2) and SU(2). The role of interaction effects in SU(N) Fermi gases as a function of $N$ were also studied in Ref. [30]. Collective excitations in SU(N) Fermi gases with tunable spin were investigated in Ref. [31]. On the other hand, the prethermalization of these systems was analyzed [32], finding that, under some conditions, the imbalanced initial state could be stabilized for a certain time. Recently, the thermodynamics of $^{87}$Sr for which $N$ can be tuned up to 10 was thoroughly analyzed in Ref. [33]. For temperatures above the super-exchange energy, the behavior of the thermodynamic quantities was found to be universal with respect to N [34]. The temperature dependence of itinerant ferromagnetism in SU(N)-symmetric Fermi gases at second order of perturbation theory has been studied recently by Huang and Cazalilla [35].

In the present work, we compare our predictions with diffusion Monte Carlo (DMC) results [9] for $S = 1/2$. We achieve a satisfactory agreement when considering hard-sphere fermions up to $k_F a_0 \simeq 1$, where the ferromagnetic transition occurs. Interestingly, in contrast to the observations in second-order perturbation theory, fermions with $S = 1/2$ exhibit a quasi-continuous phase transition into the ferromagnetic state. An important result of our study is that the critical density for itinerant ferromagnetism decreases with $N$, with values that are systematically smaller than the ones obtained in second order [20]. Our results are derived for a generic spin and thus, can be applied to SU(N) fermions. This generalization allows, for instance, the study of dilute Fermi gases of Ytterbium [26], with spin 5/2, and Strontium [27], with spin 9/2.

## 2 Methodology

We study a repulsive Fermi gas at zero temperature with spin $S$ and spin degeneracy $\nu = 2S+1$. We have used perturbation theory in our analysis, resulting in a combination of analytic results and numerical estimations as the final outcome. In the dilute gas regime, only particles with different $z$-spin component interact via a central potential $V(r)$ ($s$-wave scattering). However, since our objective is to obtain the energy up to the third order in the gas parameter, we will have to consider interactions between particles with varying $z$-spin components, involving not only the $s$-wave scattering length but also the $s$-wave effective range and the $p$-wave scattering length. We discuss the respective terms and carry out their calculations, which require the evaluation of several challenging integrals.

The first and second order terms have already been calculated in Ref. [20]. The terms coming from the $s$-wave effective range and the $p$-wave scattering length can be obtained analytically. However, the other contributions to the third-order expansion require a combination of analytical derivation and numerical integration. The number of particles in each spin channel is $N_\lambda = C_\lambda N / \nu$, with $N$ the total number of particles and $C_\lambda$ being the fraction of $\lambda$ particles (normalized to be one if the system is unpolarized, $N_\lambda = N/\nu$, $\forall \lambda$). The Fermi momentum of each species is $k_{F,\lambda} = k_F C_\lambda^{1/3}$, $k_F$ being $(6\pi^2 n/\nu)^{1/3}$. The kinetic energy is readily obtained as it corresponds to the one of the free Fermi gas,

$$\frac{T}{N} = \frac{3}{5}\epsilon_F \frac{1}{\nu} \sum_\lambda C_\lambda^{5/3}, \tag{1}$$

$\epsilon_F = \hbar^2 k_F^2 / 2m$ being the Fermi energy, and where the summation is extended to include all the $z$-spin degrees of freedom. However, obtaining the potential energy is more challenging, re-

quiring the application of perturbation theory. The formalism that we use is based on previous works [22, 36]. In essence, what is done is to calculate the Feynman diagrams that contribute to each order of the expansion, and then to substitute the interaction by the $\mathcal{K}$-matrix, which depends on the low-energy scattering parameters of the potential $V$. We have generalized this procedure considering that all the Fermi species can have any spin-channel occupation and thus including the polarization as a new variable. The potential energy can be written in terms of the $\mathcal{K}$-matrix,

$$V = \frac{\hbar^2 \Omega}{2m} \sum_{\lambda_1, \lambda_2} \int \frac{d\mathbf{p}}{(2\pi)^3} n_p \int \frac{d\mathbf{p}'}{(2\pi)^3} n_{p'} \{\mathcal{K}(\mathbf{p}, \mathbf{p}'; \mathbf{p}, \mathbf{p}') - \delta_{\lambda_1, \lambda_2} \mathcal{K}(\mathbf{p}, \mathbf{p}'; \mathbf{p}', \mathbf{p})\}, \qquad (2)$$

with $\Omega$ the volume, and $n_p$ and $n_{p'}$ the momentum distributions of the free Fermi gas. Therefore, we have to integrate over two particles ($\lambda_1, \lambda_2$) that are interacting between themselves through the $\mathcal{K}$-matrix, which brings information on the potential. The $\mathcal{K}$-matrix up to third order is written as [22, 36, 37]

$$\mathcal{K}(\mathbf{Q}, \mathbf{P}; \mathbf{Q'}, \mathbf{P'}) = 4\pi a_0 + (4\pi a_0)^2 I(Q, P) + 4\pi \mathbf{Q}^2 \frac{r_0}{2} a_0^2 + 4\pi a_1^3 \mathbf{Q} \cdot \mathbf{Q'} + (4\pi a_0)^3 I^2(Q, P)$$

$$+ 4\pi a_0 \int \frac{2 \, d\mathbf{m} d\mathbf{m'}}{(2\pi)^3} \frac{(1 - n_m)(1 - n_{m'})}{p^+ p'^2 - m^2 - m'^2} \delta(\mathbf{p} + \mathbf{p}' - \mathbf{m} - \mathbf{m}')$$

$$\times \left\{ (4\pi a_0)^2 \int \frac{2 \, d\mathbf{p}_1 d\mathbf{p}'_1}{(2\pi)^3} \frac{n_{p_1} n_{p'_1} \delta(\mathbf{p}_1 + \mathbf{p}'_1 - \mathbf{m} - \mathbf{m}')}{p_1^2 + p_1'^2 - m^2 - m'^2} \right.$$

$$+ (4\pi a_0^{(13)})(4\pi a_0^{(23)}) \sum_{\lambda_3} (2 - 3\delta_{\lambda_1, \lambda_3} - 3\delta_{\lambda_2, \lambda_3}) \int \frac{d\mathbf{p}_1 d\mathbf{m}_1}{(2\pi)^3} n_{p_1} (1 - n_{m_1})$$

$$\left. \times \left[ \frac{\delta(\mathbf{p} + \mathbf{p}_1 - \mathbf{m} - \mathbf{m}_1)}{p^2 + p_1^2 - m^2 - m_1^2} + \frac{\delta(\mathbf{p}' + \mathbf{p}_1 - \mathbf{m}' - \mathbf{m}_1)}{p'^2 + p_1^2 - m'^2 - m_1^2} \right] \right\} + O(a_0^4). \qquad (3)$$

In Eq. (3), $\mathbf{Q} = (\mathbf{p} - \mathbf{p}')/2$ and $\mathbf{P} = \mathbf{P'} = \mathbf{p} + \mathbf{p}'$ are the relative momentum and center of mass momentum, respectively. The parameters $r_0$ and $a_1$ in that equation are the $s$-wave effective range and $p$-wave scattering length, respectively. For $\mathcal{K}(\mathbf{p}, \mathbf{p}'; \mathbf{p}, \mathbf{p}')$, $\mathbf{Q'} = \mathbf{Q}$, whereas for $\mathcal{K}(\mathbf{p}, \mathbf{p}'; \mathbf{p}', \mathbf{p})$, $\mathbf{Q'} = -\mathbf{Q}$. The scattering length $a_0^{(13)}$ corresponds to the interaction between the first and third particles and $a_0^{(23)}$ to the second and third ones. When the interactions between different channels are the same, $a_0^{(13)} = a_0^{(23)} = a_0$. The term proportional to $a_0^{(13)} a_0^{(23)}$, in the expression of the $\mathcal{K}$-matrix, is a three-body interaction term. On the other hand, the function $I(Q, P)$ in Eq. (3) is defined as

$$I(Q, P) = \frac{1}{(2\pi)^3} \int 2 \, d\mathbf{q} d\mathbf{q}' \frac{1 - (1 - n_q)(1 - n_{q'})}{q^2 + q'^2 - p^2 - p'^2} \delta(\mathbf{q} + \mathbf{q}' - \mathbf{p} - \mathbf{p}'). \qquad (4)$$

Considering only the first term in the expansion of the $\mathcal{K}$-matrix (3), one gets for the potential energy (2) the well-know Hartree-Fock energy [1],

$$\left( \frac{V}{N} \right)_1 = \frac{2\epsilon_F}{3\pi} \left[ \frac{1}{\nu} \sum_{\lambda_1, \lambda_2} C_{\lambda_1} C_{\lambda_2} (1 - \delta_{\lambda_1, \lambda_2}) \right] x, \qquad (5)$$

with $x \equiv k_F a_0$ the gas parameter of the Fermi gas.

The second-order term in the gas parameter $x$ derives from the second term of the $\mathcal{K}$-matrix. This second order term reads

$$\left( \frac{V}{N} \right)_2 = \frac{\epsilon_F}{k_F^7} \left[ \frac{1}{\nu} \sum_{\lambda_1, \lambda_2} I_2(k_{F, \lambda_1}, k_{F, \lambda_2})(1 - \delta_{\lambda_1, \lambda_2}) \right] x^2, \qquad (6)$$

with

$$I_2(k_{F,\lambda_1}, k_{F,\lambda_2}) = \frac{3}{16\pi^5} \int d\mathbf{p}\, n_p \int d\mathbf{p}'\, n_{p'} \int 2\, d\mathbf{q} d\mathbf{q}' \frac{1 - (1-n_q)(1-n_{q'})}{q^2 + q'^2 - p^2 - p'^2} \delta(\mathbf{q} + \mathbf{q}' - \mathbf{p} - \mathbf{p}'). \tag{7}$$

This integral $I_2$ was already obtained in Ref. [20]. In terms of $C_{\lambda_1}$ and $C_{\lambda_2}$,

$$I_2(C_{\lambda_1}, C_{\lambda_2}) = \frac{4k_F^7}{35\pi^2} C_{\lambda_1} C_{\lambda_2} \frac{C_{\lambda_1}^{1/3} + C_{\lambda_2}^{1/3}}{2} F(y), \tag{8}$$

with

$$F(y) = \frac{1}{4}\left(15y^2 - 19y + 52 - 19y^{-1} + 15y^{-2}\right) + \frac{7}{8} y^{-2}(y-1)^4(y+3+y^{-1}) \ln\left|\frac{1-y}{1+y}\right|$$
$$- \frac{2y^4}{1+y} \ln\left|1 + \frac{1}{y}\right| - \frac{2y^{-4}}{1+y^{-1}} \ln\left|1 + y\right|, \tag{9}$$

and $y \equiv (C_{\lambda_1}/C_{\lambda_2})^{1/3}$. It is worth noticing that for $S = 1/2$ this term agrees with the Kanno result [21].

In the following subsections, we calculate the new terms, that are the ones contributing to the third order in the gas parameter.

## 2.1 S-wave effective range term

Beyond second-order, one needs to introduce additional scattering parameters other than the $s$-wave scattering length and thus the expression is no longer universal. The effective range term in the $\mathcal{K}$-matrix is $4\pi \mathbf{Q}^2 \frac{1}{2} r_0 a_0^2$. First of all, we express $\mathbf{Q}^2$ in terms of $\mathbf{p}$ and $\mathbf{p}'$,

$$\mathbf{Q}^2 = \frac{1}{4}(\mathbf{p} - \mathbf{p}')^2 = \frac{1}{4}(p^2 + p'^2 - 2pp'\cos\theta). \tag{10}$$

Then, we substitute the value of $\mathcal{K}$ in Eq. (2) and integrate it,

$$V_{r_0} = \frac{\hbar^2 \Omega}{2m} \sum_{\lambda_1,\lambda_2} \int \frac{d\mathbf{p}}{(2\pi)^3} n_p \int \frac{d\mathbf{p}'}{(2\pi)^3} n_{p'} \frac{4\pi r_0 a_0^2}{8} (p^2 + p'^2 - 2pp'\cos\theta)(1 - \delta_{\lambda_1,\lambda_2})$$
$$= \frac{\hbar^2 \Omega}{16m\pi^3} r_0 a_0^2 \sum_{\lambda_1,\lambda_2} \left[\left(\frac{k_{F,\lambda_1}^3}{3} \frac{k_{F,\lambda_2}^5}{5} + \frac{k_{F,\lambda_1}^5}{5} \frac{k_{F,\lambda_2}^3}{3}\right)(1 - \delta_{\lambda_1,\lambda_2})\right], \tag{11}$$

where the term containing $\cos\theta$ gives zero after doing the angular integration. The potential energy per particle as a function of the parameters $C_\lambda$ is finally

$$\left(\frac{V}{N}\right)_{r_0} = \epsilon_F \frac{1}{\nu} \frac{1}{10\pi} \sum_{\lambda_1,\lambda_2} \left[C_{\lambda_1} C_{\lambda_2}\left(\frac{C_{\lambda_1}^{2/3} + C_{\lambda_2}^{2/3}}{2}\right)(k_F r_0)(k_F a_0)^2(1 - \delta_{\lambda_1,\lambda_2})\right]. \tag{12}$$

## 2.2 p-wave term

The inclusion of the $p$-wave scattering length in the $\mathcal{K}$-matrix (2) introduces additional complexity as it also depends on $\mathbf{Q}'$. The two $\mathcal{K}$-matrix terms that we require differ by a minus sign: $\mathcal{K}(\mathbf{p}, \mathbf{p}'; \mathbf{p}, \mathbf{p}') = 4\pi a_1^3 \mathbf{Q}^2$, $\mathcal{K}(\mathbf{p}, \mathbf{p}'; \mathbf{p}', \mathbf{p}) = -4\pi a_1^3 \mathbf{Q}^2$. This negative sign causes the term

proportional to the Kronecker delta to change its sign, thereby introducing interaction between particles of the same spin. Although this fact may not be immediately evident, it becomes apparent once we integrate and rearrange the terms. Apart from the negative sign, the integral that needs to be performed is formally similar to the one calculated for the effective range. Hence, the potential energy per particle is

$$\left(\frac{V}{N}\right)_{a_1} = \epsilon_F \frac{1}{\nu} \frac{1}{5\pi} \sum_{\lambda_1,\lambda_2} \left[ C_{\lambda_1} C_{\lambda_2} \left( \frac{C_{\lambda_1}^{2/3} + C_{\lambda_2}^{2/3}}{2} \right) (k_F a_1)^3 (1 + \delta_{\lambda_1,\lambda_2}) \right]. \tag{13}$$

The term $(1 + \delta_{\lambda_1,\lambda_2})$ in Eq.(13) is equivalent to $2\delta_{\lambda_1,\lambda_2} + (1 - \delta_{\lambda_1,\lambda_2})$. Then, there are interaction between pairs of different spin, as in all previous terms, which are all proportional to $(1-\delta_{\lambda_1,\lambda_2})$. But now we also have a term $2\delta_{\lambda_1,\lambda_2}$, which will give rise to an extra interaction term between particles of same spin. Particularizing Eq. (13) for the latter contribution, between particles of same spin, we obtain

$$\left(\frac{V}{N}\right) = \epsilon_F \frac{1}{\nu} \frac{1}{5\pi} \sum_{\lambda_1,\lambda_2} \left[ C_{\lambda_1} C_{\lambda_2} \left( \frac{C_{\lambda_1}^{2/3} + C_{\lambda_2}^{2/3}}{2} \right) (k_F a_1)^3 (2\delta_{\lambda_1,\lambda_2}) \right]$$
$$= \epsilon_F \frac{1}{\nu} \frac{2}{5\pi} \sum_{\lambda} C_{\lambda}^{8/3} (k_F a_1)^3. \tag{14}$$

If we split Eq. (13) in two parts, one part containing the interaction between particles of different spin and another one with this new contribution, one gets

$$\left(\frac{V}{N}\right)_{a_1} = \frac{3}{5} \epsilon_F \frac{1}{\nu} \left\{ \frac{2}{3\pi} \sum_{\lambda} C_{\lambda}^{8/3} (k_F a_1)^3 + \frac{1}{3\pi} \sum_{\lambda_1,\lambda_2} \left[ C_{\lambda_1} C_{\lambda_2} \left( \frac{C_{\lambda_1}^{2/3} + C_{\lambda_2}^{2/3}}{2} \right) (k_F a_1)^3 (1 - \delta_{\lambda_1,\lambda_2}) \right] \right\}. \tag{15}$$

The primary focus of our work is the investigation of highly degenerate and extremely dilute Fermi gases, assuming that the $p$-wave interaction among particles of the same spin can be neglected. Consequently, this term (Eq. 14) will not be taken into account in the Results section.

## 2.3 3rd order terms depending on $a_0$

Due to their intricate mathematical nature, we have not been able to calculate the remaining third order terms in a fully analytical form. These terms are exclusively dependent on the $s$-wave scattering length. We used a Monte Carlo integration tool to calculate these terms.

The first term that requires consideration arises from the $(4\pi a_0)^3 I^2(Q,P)$ term in the $\mathcal{K}$-matrix expansion. Inserted in Eq. (2), one obtains

$$V_3 = \frac{\hbar^2 \Omega}{2m} \sum_{\lambda_1,\lambda_2} \int \frac{d\mathbf{p}}{(2\pi)^3} n_p \int \frac{d\mathbf{p}'}{(2\pi)^3} n_{p'} (4\pi a_0)^3 I^2(Q,P)(1 - \delta_{\lambda_1,\lambda_2}). \tag{16}$$

Rearranging the integrals, one can write a more manageable expression,

$$\left(\frac{V}{N}\right)_3 = \epsilon_F \frac{3x^3}{32\pi^7} \frac{1}{\nu} \sum_{\lambda_1,\lambda_2} \left[ (1 - \delta_{\lambda_1,\lambda_2}) E_3(C_{\lambda_1}, C_{\lambda_2}) \right], \tag{17}$$

with

$$E_3(C_{\lambda_1}, C_{\lambda_2}) = \frac{1}{k_F^8} \int d\mathbf{p} \, n_p \int d\mathbf{p}' n_{p'} \left[ \int 2 \, d\mathbf{q} d\mathbf{q}' (1 - (1 - n_q)(1 - n_{q'})) \frac{\delta(\mathbf{q} + \mathbf{q}' - \mathbf{p} - \mathbf{p}')}{q^2 + q'^2 - p^2 - p'^2} \right]^2. \tag{18}$$

Going back to the expression of the $\mathcal{K}$-matrix (2) one can see that there is another pair-like term, given by

$$
V_4 = \frac{\hbar^2 \Omega}{2m} \sum_{\lambda_1,\lambda_2} \int \frac{d\mathbf{p}}{(2\pi)^3} n_p \int \frac{d\mathbf{p}'}{(2\pi)^3} n_{p'} (1-\delta_{\lambda_1,\lambda_2})(4\pi a_0) \int \frac{2\,d\mathbf{m}d\mathbf{m}'}{(2\pi)^3} \frac{(1-n_m)(1-n_{m'})}{p^2+p'^2-m^2-m^2}
$$

$$
\times \delta(\mathbf{p}+\mathbf{p}'-\mathbf{m}-\mathbf{m}')(4\pi a_0)^2 \int \frac{2\,d\mathbf{p}_1 d\mathbf{p}_1'}{(2\pi)^3} n_{p_1} n_{p_1'} \frac{\delta(\mathbf{p}_1+\mathbf{p}_1'-\mathbf{m}-\mathbf{m}')}{p_1^2+p_1'^2-m^2-m'^2}. \tag{19}
$$

Observing that $\mathbf{p}$ and $\mathbf{p}_1$ run over the same values, and the same happens for $\mathbf{p}'$ and $\mathbf{p}_1'$, we can interchange the integrals and rewrite the whole expression as

$$
\left(\frac{V}{N}\right)_4 = \epsilon_F \frac{3x^3}{32\pi^7} \frac{1}{\nu} \sum_{\lambda_1,\lambda_2} \left[(1-\delta_{\lambda_1,\lambda_2})E_4(C_{\lambda_1},C_{\lambda_2})\right], \tag{20}
$$

with

$$
E_4(C_{\lambda_1},C_{\lambda_2}) = \frac{1}{k_F^8} \int d\mathbf{m}(1-n_m) \int d\mathbf{m}'(1-n_{m'}) \left[\int 2\,d\mathbf{p}d\mathbf{p}' n_p n_{p'} \frac{\delta(\mathbf{p}+\mathbf{p}'-\mathbf{m}-\mathbf{m}')}{p^2+p'^2-m^2-m^2}\right]^2. \tag{21}
$$

Finally, the last term in Eq. (2) contains the interaction between three particles. At third order, this is the only three-body interacting term. Its expression is more involved than the previous ones,

$$
V_5 = \frac{\hbar^2 \Omega}{2m} \sum_{\lambda_1,\lambda_2} \int \frac{d\mathbf{p}}{(2\pi)^3} n_p \int \frac{d\mathbf{p}'}{(2\pi)^3} n_{p'} (1-\delta_{\lambda_1,\lambda_2})(4\pi a_0)^3
$$

$$
\times \int \frac{2\,d\mathbf{m}d\mathbf{m}'}{(2\pi)^3} (1-n_m)(1-n_{m'}) \frac{\delta(\mathbf{p}+\mathbf{p}'-\mathbf{m}-\mathbf{m}')}{p^2+p'^2-m^2-m^2}
$$

$$
\times \sum_{\lambda_3} (2-3\delta_{\lambda_1,\lambda_3}-3\delta_{\lambda_2,\lambda_3}) \int \frac{d\mathbf{p}_1 d\mathbf{m}_1}{(2\pi)^3} n_{p_1}(1-n_{m_1})
$$

$$
\times \left[\frac{\delta(\mathbf{p}+\mathbf{p}_1-\mathbf{m}-\mathbf{m}_1)}{p^2+p_1^2-m^2-m_1^2} + \frac{\delta(\mathbf{p}'+\mathbf{p}_1-\mathbf{m}'-\mathbf{m}_1)}{p'^2+p_1^2-m'^2-m_1^2}\right]. \tag{22}
$$

After rearranging, we can write it in a more compact form

$$
\left(\frac{V}{N}\right)_5 = \epsilon_F \frac{3x^3}{32\pi^7} \frac{1}{\nu} \sum_{\lambda_1,\lambda_2,\lambda_3} \left[(1-\delta_{\lambda_1,\lambda_2})(2-3\delta_{\lambda_1,\lambda_3}-3\delta_{\lambda_2,\lambda_3})E_5(C_{\lambda_1},C_{\lambda_2},C_{\lambda_3})\right], \tag{23}
$$

with

$$
E_5(C_{\lambda_1},C_{\lambda_2},C_{\lambda_3}) = \frac{1}{2k_F^8} \left\{ \int d\mathbf{p} n_p \int d\mathbf{m}(1-n_m) \left[\int 2d\mathbf{m}'d\mathbf{p}'(1-n_{m'})n_{p'}\right.\right.
$$

$$
\left.\times \frac{\delta(\mathbf{p}+\mathbf{p}'-\mathbf{m}-\mathbf{m}')}{p^2+p'^2-m^2-m'^2}\right]\left[\int 2d\mathbf{m}_1 d\mathbf{p}_1(1-n_{m_1})n_{p_1} \frac{\delta(\mathbf{p}+\mathbf{p}_1-\mathbf{m}-\mathbf{m}_1)}{p^2+p_1^2-m^2-m_1^2}\right]
$$

$$
+ \int d\mathbf{p}' n_{p'} \int d\mathbf{m}'(1-n_{m'}) \left[\int 2d\mathbf{m}d\mathbf{p}(1-n_m)n_p \frac{\delta(\mathbf{p}+\mathbf{p}'-\mathbf{m}-\mathbf{m}')}{p^2+p'^2-m^2-m'^2}\right]
$$

$$
\left.\times \left[\int 2d\mathbf{m}_1 d\mathbf{p}_1(1-n_{m_1})n_{p_1} \frac{\delta(\mathbf{p}'+\mathbf{p}_1-\mathbf{m}'-\mathbf{m}_1)}{p'^2+p_1^2-m'^2-m_1^2}\right]\right\}. \tag{24}
$$

The integrals $E_3$, $E_4$, and $E_5$ were already calculated previously for a non-polarized gas and $S = 1/2$ [16–19, 36, 38]. In order to calculate numerically the integrals for spins greater than 1/2, we have made the assumption that as the concentration of one species increases, all the other species diminish in the same manner. This particular configuration minimizes the energy when the total number of particles remains constant [20]. Under these conditions, the concentrations $C_\lambda$ for a given polarization $P$ are

$$C_+ = 1 + |P|(\nu - 1), \tag{25}$$
$$C_{\lambda \neq +} = 1 - |P|, \tag{26}$$

with subindex + standing for the spin state with the largest population. The explicit way in which these integrals are calculated can be found in the Appendices. We have computed these integrals for different values of the polarization and a range of spin values. More precisely, we have calculated 201 points between $P = 0$ and $P = 1$, with $10^8$ sampling points using an accurate adaptive Monte Carlo integration [39, 40].

## 2.4 Energy up to third order

Collecting the different terms discussed in previous subsections, we can write a final expression for the energy of a Fermi gas up to third order in the gas parameter $k_F a_0$, and for any spin degeneracy and polarization,

$$
\begin{aligned}
\frac{E}{N} = \frac{3}{5}\epsilon_F \Bigg\{ & \frac{1}{\nu}\sum_\lambda C_\lambda^{5/3} + \frac{1}{\nu}\sum_\lambda \frac{2}{3\pi} C_\lambda^{8/3} (k_F a_1)^3 + \frac{5}{3\nu}\sum_{\lambda_1,\lambda_2}\Bigg[\Bigg(\frac{2}{3\pi}(k_F a_0)C_{\lambda_1}C_{\lambda_2} \\
& + \frac{4}{35\pi^2}C_{\lambda_1}C_{\lambda_2}\frac{C_{\lambda_1}^{1/3}+C_{\lambda_2}^{1/3}}{2}F(y)(k_F a_0)^2 + \frac{1}{10\pi}C_{\lambda_1}C_{\lambda_2}\Bigg(\frac{C_{\lambda_1}^{2/3}+C_{\lambda_2}^{2/3}}{2}\Bigg)\Bigg[\frac{r_0}{a_0}+2\frac{a_1^3}{a_0^3}\Bigg](k_F a_0)^3 \\
& + \frac{3}{32\pi^7}\Big[E_3 + E_4 + \sum_{\lambda_3}((2-3\delta_{\lambda_1,\lambda_3}-3\delta_{\lambda_2,\lambda_3})E_5)\Big](k_F a_0)^3\Bigg)(1-\delta_{\lambda_1,\lambda_2})\Bigg]\Bigg\}. \tag{27}
\end{aligned}
$$

# 3 Results

In this section, we discuss the main results of SU(N) Fermi gases using the framework established in the previous section. Unlike the second order analysis, we now introduce two additional scattering parameters that characterize the interaction: the $s$-wave effective range ($r_0$) and the $p$-wave scattering length ($a_1$). This implies that the specific potential model has an influence on the derived results beyond the simple dependence on the $s$-wave scattering length. Considering our focus on a repulsive gas, unless otherwise stated, we will assume a hard-sphere potential ($r_0 = 2a_0/3$, $a_1 = a_0$), as it provides a somewhat more universal model.

Having a perturbative prediction at our disposal, we can make a comparison between our results and the existing quantum Monte Carlo data. We compare two sets of values in Fig. 1. The black points are diffusion Monte Carlo (DMC) data for spin 1/2 [9]. This set does not include $P$-wave scattering terms between particles with the same $z$-spin component. The brown points in the same figure are DMC data by Bertaina et al. [41] which include intra-species interaction. The blue and purple lines are our theoretical predictions for these two cases, respectively. Both results correspond to non-polarized gases. As we can see, the energy is higher when the intra-species interaction is considered. Moreover, although it is not shown here, intra-species interaction in the fully-polarized gas make the energy increase with the density [41, 42]. In the rest of our results, we do not include this contribution. The orange line

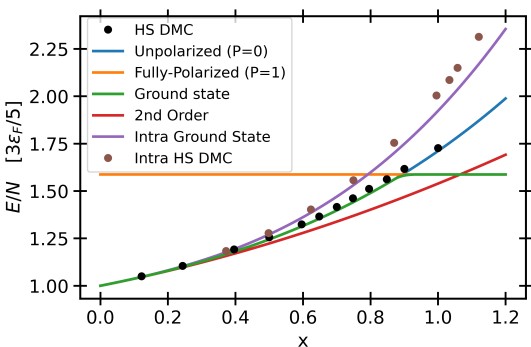

Figure 1: Energy for $S = 1/2$ as a function of the gas parameter $x = k_F a_0$. We plot three models: in red, the second-order model; in green (and blue), the third-order one; and in purple, the third-order one with $P$-wave intra-species interaction. The black and brown points are DMC results from Ref. [9] and [41] respectively. The potential used is hard-spheres ($r_0 = 2 a_0/3$, $a_1 = a_0$).

corresponds to the fully-polarized gas, the green one stands for the configuration of minimum energy, that is, at each value of the density, we select the polarization that minimizes the energy. And finally, the red line corresponds to the second-order energy (universal expansion) to show the differences with the third-order expansion. We can see how the second-order and the third-order expansions reproduce the same energy for values of $k_F a_0 \lesssim 0.4$. Beyond that, the difference intensifies with increasing density [14]. Concerning the DMC points, although they are upper-bounds to the exact energy due to the sign problem, they fit pretty well the third-order curves.

In Fig. 2, we analyze the dependence of the energy with the gas parameter for a soft-sphere potential. The points are DMC data from Ref. [9]. Here, the scattering parameters are $r_0 = 0.424 a_0$ and $a_1 = 1.1333 a_0$. We see that the third-order energies reproduce accurately the DMC data up to a value of $k_F a_0 \simeq 0.8$ and, after that, Eq. 27 starts to depart from the DMC energies. The reason is that our perturbative expansion works for $k_F a_0 < 1$, but also for $k_F a_1 < 1$. As now $a_1$ is larger than $a_0$, the range of convergence has been reduced.

Equation 27 gives the energy as a function of the gas parameter but importantly also as a function of the polarization. In Fig. 3, we show our results for different gas parameters. One can see that, until a gas parameter of $k_F a_0 = 0.85$, the polarization that minimizes the energy

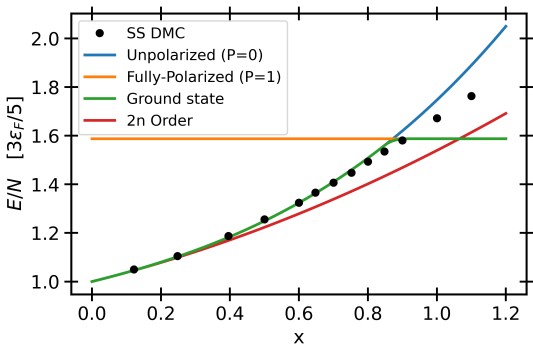

Figure 2: Energy for $S = 1/2$ as a function of the gas parameter. We plot two models: in green, the third-order model; in red, the second-order one. The points are DMC results from Ref. [9]. The potential used is a soft-sphere model with $r_0 = 0.424 a_0$, and $a_1 = 1.1333 a_0$.

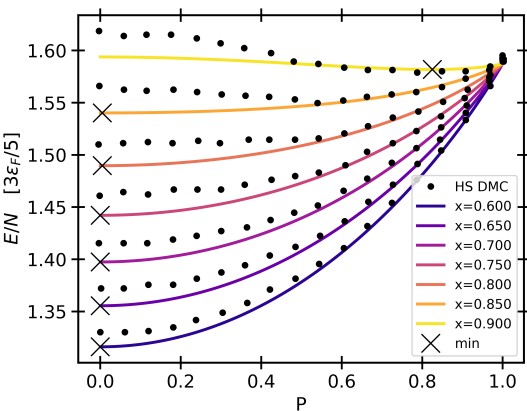

Figure 3: Energy per particle of a $S = 1/2$ Fermi gas as a function of the polarization and for different values of the gas parameter. The solid points stand for DMC results [9] and the crosses to our prediction for the polarization giving the minimum energy. The potential is a hard-sphere one.

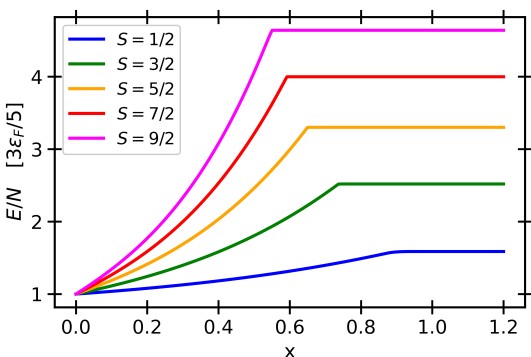

Figure 4: Energy per particle as a function of the gas parameter for spins $S = 1/2, 3/2, 5/2, 7/2$, and $9/2$.

(shown as crosses in the figure) is zero. However, at $k_F a_0 = 0.9$, the minimum of the energy moves to a larger polarization. As it is clear in the figure, the DMC points [9] fit progressively better to our results when the polarization increases. A possible explanation for this behavior could be the different quality of the model nodal surface used in DMC calculations: it is well known that the plane-waves Slater determinant used in those DMC calculations is better for a polarized than for an unpolarized Fermi gas [43, 44].

In Fig. (4), we show the energy as a function of the gas parameter for five spin values $S = 1/2, 3/2, 5/2, 7/2$, and $9/2$. It is worth noticing that for $S > 1/2$ there are not available DMC to compare with. The results reported in the figure correspond to a hard-sphere interaction. As one can see, the third-order model predicts a ferromagnetic phase transition for the five spin values, since the curve becomes flat after a certain characteristic $x$. As observed also in second order [20], the ferromagnetic transition occurs at lower values of the gas parameter when $S$ increases. For a same spin value, the critical density decreases when third-order terms are introduced in the expansion.

In order to gain a deeper understanding of the phase transition, we plot in Fig. 5 the order parameter (in this instance, the polarization) as a function of the gas parameter $x$. The polarization we plot is the one that minimizes the energy at a given $x$. Indeed, as previously commented, all gases evolve from a non-polarized to a fully-polarized state. However, there

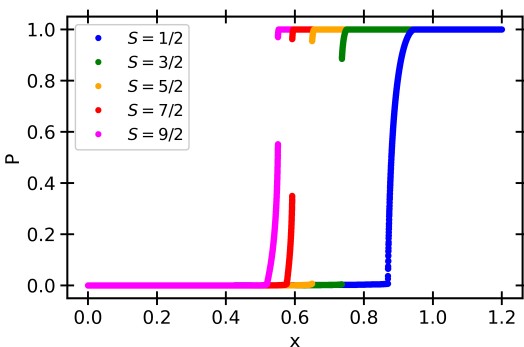

Figure 5: Polarization as a function of the gas parameter for spins $S = 1/2, 3/2, 5/2, 7/2$, and $9/2$.

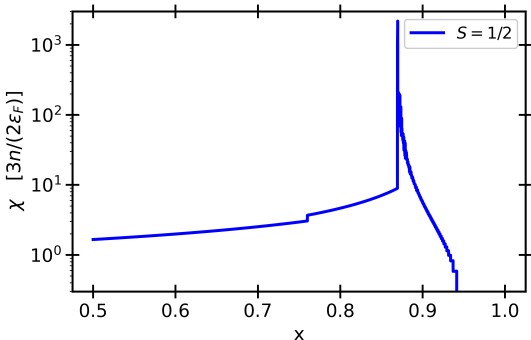

Figure 6: Magnetic susceptibility as a function of the gas parameter for spin $S = 1/2$. The y-scale is semi-log.

are notable distinctions among them. For spin $1/2$, the transition is quasi-continuous, this fact contrasts with the second-order result reported in Ref. [20] where the transition for spin $1/2$ was first order and had a polarization jump of 0.545. By quasi-continuous, we mean that the transition could be discontinuous, but with a tiny jump. Apparently our results show a continuous transition, but at third order we no longer have a fully analytical expression and, hence, our prediction has the limits of our numerical accuracy. For spin $3/2$ and $5/2$, we have partial discontinuous transitions, as there is a polarization jump, but it does not reach $P = 1$, hence, the label 'partial'. If it reached $P = 1$ directly, it would be a total discontinuous transition. Astonishingly, the gases with the last two largest spins, $S = 7/2$ and $9/2$, experience a continuous transition when increasing the density, but the continuous transition is truncated because a discontinuous transition occurs before $P$ reaches 1.

Fundamental information on the magnetic properties of the Fermi gases is contained in the magnetic susceptibility $\chi$,

$$\frac{1}{\chi} = \frac{1}{n}\left(\frac{\partial^2 (E/N)}{\partial P^2}\right)_x . \tag{28}$$

Third-order results of $\chi$ for spin $1/2$ are shown in Fig. 6, and the ones for larger spins in Fig. 7. We split the results in two figures because, as spin $1/2$ suffers a quasi-continuous transition, the magnetic susceptibility diverges. We recall that, for second-order phase transitions, the susceptibility must diverge. Notice, however, that we obtain a very large value ($\sim 2e3$) at $k_F a_0 = 0,85$ and not a real divergence due to our finite numerical precision. If the accuracy is improved, the critical value increases. The behavior of the magnetic susceptibility for spin $1/2$ confirms the behavior of the polarization shown in Fig. 5.

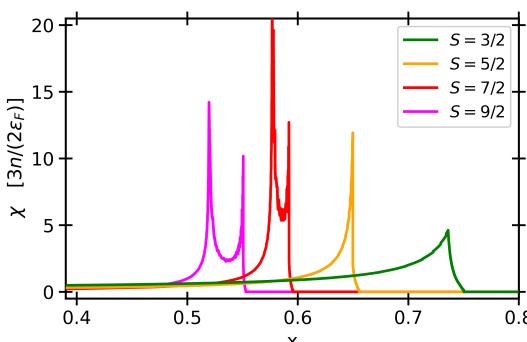

Figure 7: Magnetic susceptibility as a function of the gas parameter for spins $S = 3/2, 5/2, 7/2$, and $9/2$.

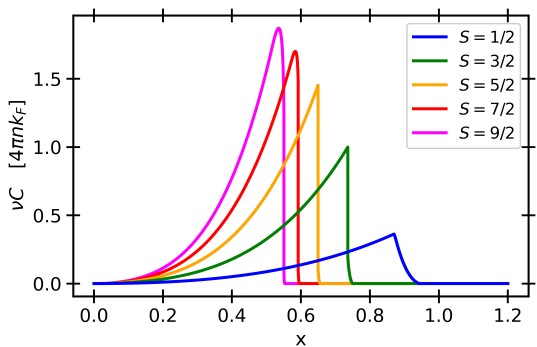

Figure 8: Tan's constant as a function of the gas parameter for spins $S = 1/2, 3/2, 5/2, 7/2$, and $9/2$).

The other $\chi$ results for larger spins (Fig. 7) exhibit the behavior we have predicted above. For spins $3/2$ and $5/2$, we only have a finite peak, telling us that there is a discontinuous transition. For spins $7/2$ and $9/2$, we see the singular double transition (two peaks) that we have mentioned before. With increasing density, the first peak corresponds to the truncated continuous transition, and the second peak to the latter first-order phase transition. The two peaks point to the existence of an intermediate phase between the non-polarized phase and the fully-polarized one. This phase would be located around the bottom that lies between peaks in Fig. 7. And, according to Fig. 5, this intermediate phase would have a partial polarization, hence, the Fermi gas would exhibit some kind of magnetic ordering. The rich phase diagram that appears in SU(N) Fermi systems have also been pointed out in Ref. [45].

The Tan's constant,

$$ C = \frac{8\pi m a_0^2}{\nu \hbar^2} \frac{N}{V} \frac{\partial(E/N)}{\partial a_0}, \tag{29} $$

is a very good tool to locate the density at which the itinerant ferromagnetism transition occurs. It is so, because right before the transition, the Tan's constant value is maximum (see Fig. 8). In terms of the gas parameter, it is

$$ C = \frac{4\pi n k_F}{\nu} \frac{x^2}{\epsilon_F} \frac{\partial(E/N)}{\partial x}. \tag{30} $$

The Tan's constant results for five values of the spin are shown in Fig. 8. One can see again that the transition happens at lower densities when the degeneracy increases.

As we are now beyond universality, it is interesting to explore the role of the *s*-wave effective range and *p*-wave scattering length in the location of the critical density. To reduce the

dimensionality of the problem, we have analyzed the particular case of $a_1 = a_0$ and spin 1/2. In Fig. 9, we plot the Tan's constant by changing only the effective range. As one can see, the critical density changes significantly with $r_0$. Interestingly, increasing $r_0$ the ferromagnetic transition moves to lower densities making it somehow more accessible in experiments.

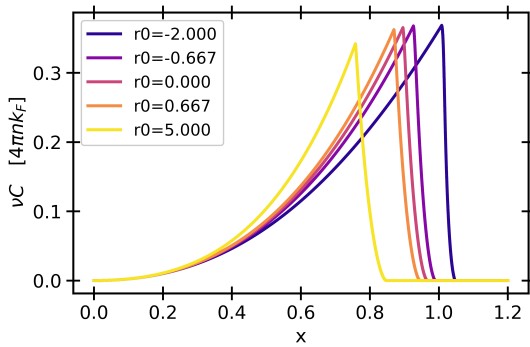

Figure 9: Tan's constant as a function of the gas parameter for spin 1/2. The $P$-wave scattering length is $a_1 = a_0$ as for the hard-sphere interaction. The effective range varies from $-2\,a_0$ to $5\,a_0$.

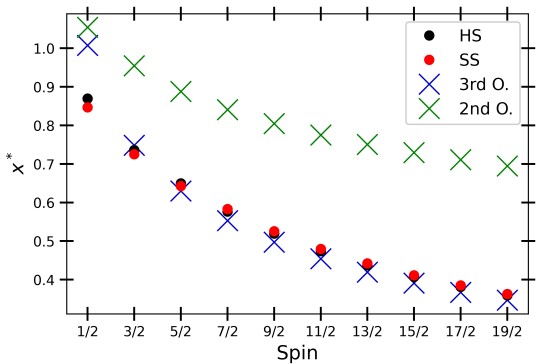

Figure 10: Critical gas parameter as a function of the spin value. The black and red points stand for the hard-sphere ($r_0 = 2a_0/3$, $a_1 = a_0$) and soft-sphere potentials ($r_0 = 0.424\,a_0$, $a_1 = 1.1333\,a_0$), respectively. The green crosses are second-order results. The blue crosses are third-order results including only the terms that depend on $a_0$.

As commented before, the ferromagnetic phase transition happens at lower densities if we increase the spin of the gas. In Fig. 10, we show the critical gas parameter as a function of the spin value, up to spin 19/2. The inclusion of third-order terms reduces the critical $x^*$, with an effect that increases with the spin due to the enhanced role of interactions between different spin channels. We report the critical gas parameters obtained using both the hard-sphere and soft-sphere potentials. The soft-sphere potential is the same we used in Fig. 2, this means $r_0 = 0.424\,a_0$ and $a_1 = 1.1333\,a_0$. The difference between these two potentials is almost imperceptible except somehow for $S = 1/2$ where the critical value for soft-spheres is slightly below the hard-sphere one. It is interesting to explore what is the relevance of the effective range and $p$-wave contributions with respect to the one coming from the third-order terms that depend only on $a_0$ on the critical value. To this end, we show in Fig. 10 third-order results depending only on $a_0$ (blue crosses). As one can see, the role of $r_0$ and $a_1$ is small for $S > 1/2$ but substantial for the particular case $S = 1/2$. Neglecting the effect of scattering

parameters others than $a_0$, we estimate that the transition gas parameter is 0.65 for Ytterbium ($S = 5/2$), and 0.53 for Strontium ($S = 9/2$).

## 4 Conclusions

To summarize, we have derived the expression for the energy of a repulsive SU(N) Fermi gas, incorporating terms up to third order in the gas parameter and in terms of the spin-channel occupations. We have extended up to third order our second-order expansion [20], written also in terms of the spin polarization. We have included the analytic terms dependent on the $s$-wave effective range and the $p$-wave scattering length, while the remaining third-order terms have been computed numerically. Although these numerical terms introduce some level of uncertainty, we have managed to minimize it to the best of our abilities. The uncertainty in the values of $P$ are typically 0.005, and for the energy are about $10^{-6}$.

In order to study the Fermi gas, and to reduce the number of variables to explore, we have selected the occupational configuration that minimizes the energy. At $P = 0$, all species are equally occupied, and as $P$ increases, one species increases while all others decrease in the same manner. However, we point out that our formalism can be applied to any occupational configuration. One just needs to find the new expressions for the fractions of $\lambda$ particles $C_\lambda$. In fact, in several experiments, the species' occupation can be tuned or controlled almost at will [31, 33] and there are works that have dealt with imbalanced systems [32]. However, we point out that all these other configurations represent excited states with higher energy compared to the configuration we have chosen. Consequently, if a Fermi system is capable of thermalization, it will converge to the behavior predicted in our work. Notice that adopting a different occupational configuration would yield distinct critical values for the gas parameter $x^*$.

The third-order expression allows to reproduce and test numerical values obtained with diffusion Monte Carlo. In the case of itinerant ferromagnetism, where three scattering parameters come into play, the existence of the phase transition could even not exist. This is not the case for the interactions analyzed in the present work: for a hard-sphere potential and using the third-order energies, the spin 1/2 Fermi gas exhibits a quasi-continuous magnetic phase transition, in contrast with the first-order phase transition predicted by the second-order approximation (universal expansion) [20]. According to Ref. [46], the order of the $S = 1/2$ ferromagnetic transition is always first-order because the sign of the term $P^4 \ln P$ in the Landau expansion must be positive. However, and within our numerical precision, this term is in fact negative at third-order and thus, the transition becomes quasi-continuous. The same change of sign is observed in [47] where the authors apply a resummation method. For higher spins values, we have different situations. Spins 3/2 and 5/2 exhibit a first-order transition. However, gases with larger spins show a double transition. First, a continuous transition happens, but, before reaching $P = 1$, a discontinuous transition truncates it.

Importantly, the inclusion of the third-order terms significantly reduces the critical value of the gas parameter at which the ferromagnetic transition is expected. For spin 5/2, it occurs at $k_F a_0 = 0.65$, and for spin 9/2, at $k_F a_0 = 0.53$. This finding reinforces the notion that the observation of itinerant ferromagnetism may be more favorable when working with highly degenerate gases like Yb [26] and Sr [27]. Beyond the second-order approximation, the energy ceases to be universal in terms of the gas parameters because the $s$-wave effective range and $p$-wave scattering length come into play [22]. However, for larger spins, the region of interest (where the transition occurs) tends to be at lower densities, as mentioned earlier, leading these systems toward a new form of universality.

## Acknowledgments

Stimulating discussions with Miguel Cazalilla and Chen-How Huang are aknowledged.

**Funding information** We acknowledge financial support from MCIN/AEI/10.13039/ 501100011033(Spain) Grant No. PID2020-113565GB-C21 and from AGAUR- Generalitat de Catalunya Grant N0. 2021-SGR-01411.

## A  Numerical integration of third-order terms

We have three terms contributing at third order of the perturbative series which are numerically integrated. The first two terms involve interaction between two particles, however, the third one is a three-body interaction term. In the following subsections, each one of these terms is analyzed.

### A.1  Term E3

The expression of the $E_3$ term is

$$\frac{E}{N} = \epsilon_F \frac{3x^3}{32\pi^7} \frac{1}{\nu} \sum_{\lambda_1, \lambda_2} (1 - \delta_{\lambda_1, \lambda_2}) E_3(C_{\lambda_1}, C_{\lambda_2}), \tag{A.1}$$

with

$$E_3(C_{\lambda_1}, C_{\lambda_2}) = \frac{1}{k_F^8} \int d\mathbf{p}\, n_p \int d\mathbf{p}'\, n_{p'} \left[ \int 2\, d\mathbf{q} d\mathbf{q}' \frac{1 - (1 - n_q)(1 - n_{q'})}{q^2 + q'^2 - p^2 - p'^2} \delta(\mathbf{q} + \mathbf{q}' - \mathbf{p} - \mathbf{p}') \right]^2. \tag{A.2}$$

We recall that $\mathbf{p}$ and $\mathbf{q}$ run over $k_{F,\lambda_1}$, while $\mathbf{p}'$ and $\mathbf{q}'$ run over $k_{F,\lambda_2}$. If we expand the numerator, where the occupation functions are, we see that we can split the terms: two containing $n_q$ and the other one with $n_q n_{q'}$. This is telling us the volume where we have to integrate. For the first two terms, $n_q$ gives an sphere, and, for the third one, $n_q n_{q'}$ gives a more complicated volume, as it is the intersection of two spheres. Therefore, we have to solve two integrals, the one having the volume of a sphere (Sphere-Integral -SI-), and the other one with the intersection (Sphere-Intersection-Integral -SII-). Then,

$$E_3(C_{\lambda_1}, C_{\lambda_2}) = \frac{1}{k_F^8} \int d\mathbf{p}\, n_p \int d\mathbf{p}'\, n_{p'} \left[ SI(p, p', k_{F,\lambda_1}) + SI(p, p', k_{F,\lambda_2}) - SII(p, p', k_{F,\lambda_1}, k_{F,\lambda_2}) \right]^2. \tag{A.3}$$

The SI terms can be integrated in the following way,

$$
\begin{aligned}
SI(p, p', k_{F,\lambda}) &= \int 2d\mathbf{q}d\mathbf{q}' \frac{n_q}{q^2 + q'^2 - p^2 - p'^2} \delta(\mathbf{q} + \mathbf{q}' - \mathbf{p} - \mathbf{p}') \\
&= \int d\mathbf{q} \frac{n_q}{q^2 - \mathbf{q}\cdot(\mathbf{p}+\mathbf{p}') + \mathbf{p}\cdot\mathbf{p}'} \\
&= \int_0^{k_{F,\lambda}} q^2 dq \int_0^\pi \sin\theta d\theta \frac{2\pi}{q^2 - q|\mathbf{p}+\mathbf{p}'| + \mathbf{p}\cdot\mathbf{p}'} \\
&= 2\pi \int_0^{k_{F,\lambda}} q^2 dq \frac{1}{q|\mathbf{p}+\mathbf{p}'|} \ln\left[\frac{q^2 + q|\mathbf{p}+\mathbf{p}'| + \mathbf{p}\cdot\mathbf{p}'}{q^2 - q|\mathbf{p}+\mathbf{p}'| + \mathbf{p}\cdot\mathbf{p}'}\right] \\
&= \frac{2\pi}{|\mathbf{p}+\mathbf{p}'|}\left\{\left(\frac{k_{F,\lambda}^2}{2} - \frac{p^2 + p'^2}{4}\right)\ln\left[\frac{k_{F,\lambda}^2 + k_{F,\lambda}|\mathbf{p}+\mathbf{p}'| + \mathbf{p}\cdot\mathbf{p}'}{k_{F,\lambda}^2 - k_{F,\lambda}|\mathbf{p}+\mathbf{p}'| + \mathbf{p}\cdot\mathbf{p}'}\right]\right. \\
&\quad \left. - \frac{|\mathbf{p}+\mathbf{p}'||\mathbf{p}-\mathbf{p}'|}{4}\ln\left[\frac{k_{F,\lambda}^2 + k_{F,\lambda}|\mathbf{p}-\mathbf{p}'| - \mathbf{p}\cdot\mathbf{p}'}{k_{F,\lambda}^2 - k_{F,\lambda}|\mathbf{p}-\mathbf{p}'| - \mathbf{p}\cdot\mathbf{p}'}\right] + k_{F,\lambda}|\mathbf{p}+\mathbf{p}'|\right\}.
\end{aligned}
\tag{A.4}
$$

Now, we perform the following change of variable that will be useful later on,

$$
\begin{aligned}
\mathbf{P} = \mathbf{p} + \mathbf{p}', &\qquad\qquad \mathbf{p} = \frac{\mathbf{P}}{2} + \mathbf{Q}, \\
&\Rightarrow \\
\mathbf{Q} = \frac{\mathbf{p} - \mathbf{p}'}{2} &\qquad\qquad \mathbf{p}' = \frac{\mathbf{P}}{2} - \mathbf{Q}.
\end{aligned}
\tag{A.5}
$$

With these new variables the sphere integral can be rewritten as follows,

$$
\begin{aligned}
SI(P, Q, k_{F,\lambda}) = \frac{2\pi}{P}\left\{\left(\frac{k_{F,\lambda}^2}{2} - \frac{1}{2}\left(\frac{P^2}{4} + Q^2\right)\right)\ln\left|\frac{\left(k_{F,\lambda} + \frac{P}{2}\right)^2 - Q^2}{\left(k_{F,\lambda} - \frac{P}{2}\right)^2 - Q^2}\right|\right. \\
\left. - \frac{PQ}{2}\ln\left|\frac{\left(k_{F,\lambda} + Q\right)^2 - \frac{P^2}{4}}{\left(k_{F,\lambda} - Q\right)^2 - \frac{P^2}{4}}\right| + k_{F,\lambda}P\right\}.
\end{aligned}
\tag{A.6}
$$

When $P$ is zero, $SI$ could diverge, however, taking the limit of $P$ going to zero, one can see that the result is finite. This function when $P \to 0$ is

$$
SI(0, Q, k_{F,\lambda}) = 2\pi\left[2k_{F,\lambda} - Q\ln\left|\frac{k_{F,\lambda} + Q}{k_{F,\lambda} - Q}\right|\right].
\tag{A.7}
$$

Now, we analyze the $SII$ term. We first apply the change of variables introduced above, and then we integrate over $P$. Then,

$$
\begin{aligned}
SII(p, p', k_{F,\lambda_1}, k_{F,\lambda_2}) &= \int 2d\mathbf{q}d\mathbf{q}' \frac{n_q n_{q'}}{q^2 + q'^2 - p^2 - p'^2} \delta(\mathbf{q} + \mathbf{q}' - \mathbf{p} - \mathbf{p}') \\
&= \int d\mathbf{Q}d\mathbf{P} \frac{n_{P/2+Q} n_{P/2-Q}}{Q^2 + P^2/4 - 2(p^2 + p'^2)/4} \delta(\mathbf{P} - \mathbf{p} - \mathbf{p}') \\
&= \int d\mathbf{Q} \frac{n_{P/2+Q} n_{P/2-Q}}{Q^2 - \frac{(p-p')^2}{4}}.
\end{aligned}
\tag{A.8}
$$

Table 1: Different types of intersection between two spheres depending on the distance between centers.

| Condition | Intersection |
|---|---|
| $0 \leq P < |k_{F,\lambda_1} - k_{F,\lambda_2}|$ | The small sphere |
| $P = |k_{F,\lambda_1} - k_{F,\lambda_2}|$ | The small sphere (Inner tangency) |
| $|k_{F,\lambda_1} - k_{F,\lambda_2}| < P < (k_{F,\lambda_1} + k_{F,\lambda_2})$ | Two spherical caps |
| $P = (k_{F,\lambda_1} + k_{F,\lambda_2})$ | A point (Outer tangency) |
| $P > (k_{F,\lambda_1} + k_{F,\lambda_2})$ | Null intersection |

The term $n_{P/2+Q} n_{P/2-Q}$ is telling us the volume that we have to integrate. This comes from the intersection of spheres

$$I: \quad \left(Q_x + \frac{P_x}{2}\right)^2 + \left(Q_y + \frac{P_y}{2}\right)^2 + \left(Q_z + \frac{P_z}{2}\right)^2 \leq k_{F,\lambda_1}^2, \tag{A.9}$$

$$II: \quad \left(Q_x - \frac{P_x}{2}\right)^2 + \left(Q_y - \frac{P_y}{2}\right)^2 + \left(Q_z - \frac{P_z}{2}\right)^2 \leq k_{F,\lambda_2}^2. \tag{A.10}$$

As we see, we have two spheres with different radii. The first one is centered at $\left(\frac{-P_x}{2}, \frac{-P_y}{2}, \frac{-P_z}{2}\right)$ with radius $k_{F,\lambda_1}$; and the second one is centered at $\left(\frac{P_x}{2}, \frac{P_y}{2}, \frac{P_z}{2}\right)$ with radius $k_{F,\lambda_2}$. And $\mathbf{Q}$ must satisfy both constraints, that is, we have to integrate the intersection of the two spheres. Depending on the distance between the spheres we will have different kind of intersections. The distance between the two centers is just $P$. If $P$ is larger than the sum of both radius, then there is no intersection and the integral will be zero. If $P$ is lower than the difference of radii, then the small sphere will be completely inside the big sphere, therefore the intersection will be the small sphere and the integral will be exactly $SI$ because the volume will be just an sphere. And finally, when $P$ is smaller than the sum of both radii and larger than their difference, the intersection corresponds to the sum of two spherical caps, one coming from each sphere. This explanation is summarized in Table 1 and plotted in Fig. 11.

Therefore, one only needs to analyze the case of spherical caps, $|k_{F,\lambda_1} - k_{F,\lambda_2}| < P < (k_{F,\lambda_1} + k_{F,\lambda_2})$. As the coordinates of $P$ can take a large set of values and thus we can have many distributions of the spheres in the space, we will perform a 3D-rotation in order to set our system in a vertical position regardless of the initial configuration. More specifically, the sphere with radius $k_{F,\lambda_2}$ will be at the top, and the one with radius $k_{F,\lambda_1}$ at the bottom.

We want the following transformations for the rotation: $\left(\frac{-P_x}{2}, \frac{-P_y}{2}, \frac{-P_z}{2}\right) \Rightarrow \left(0, 0, \frac{-P}{2}\right)$ and $\left(\frac{P_x}{2}, \frac{P_y}{2}, \frac{P_z}{2}\right) \Rightarrow \left(0, 0, \frac{P}{2}\right)$, but in fact they are the same transformation because the minus sign makes no difference, and also the one half multiplying can be taken out. Under this rotation, the change of variables is

$$Q_x = \frac{P_x}{\sqrt{P_x^2 + P_y^2}} \frac{P_z}{P} Q_{\alpha x} - \frac{P_y}{\sqrt{P_x^2 + P_y^2}} Q_{\alpha y} + \frac{P_x}{P} Q_{\alpha z}, \tag{A.11}$$

$$Q_y = \frac{P_y}{\sqrt{P_x^2 + P_y^2}} \frac{P_z}{P} Q_{\alpha x} + \frac{P_x}{\sqrt{P_x^2 + P_y^2}} Q_{\alpha y} + \frac{P_y}{P} Q_{\alpha z}, \tag{A.12}$$

$$Q_z = -\frac{\sqrt{P_x^2 + P_y^2}}{P} Q_{\alpha x} + \frac{P_z}{P} Q_{\alpha z}. \tag{A.13}$$

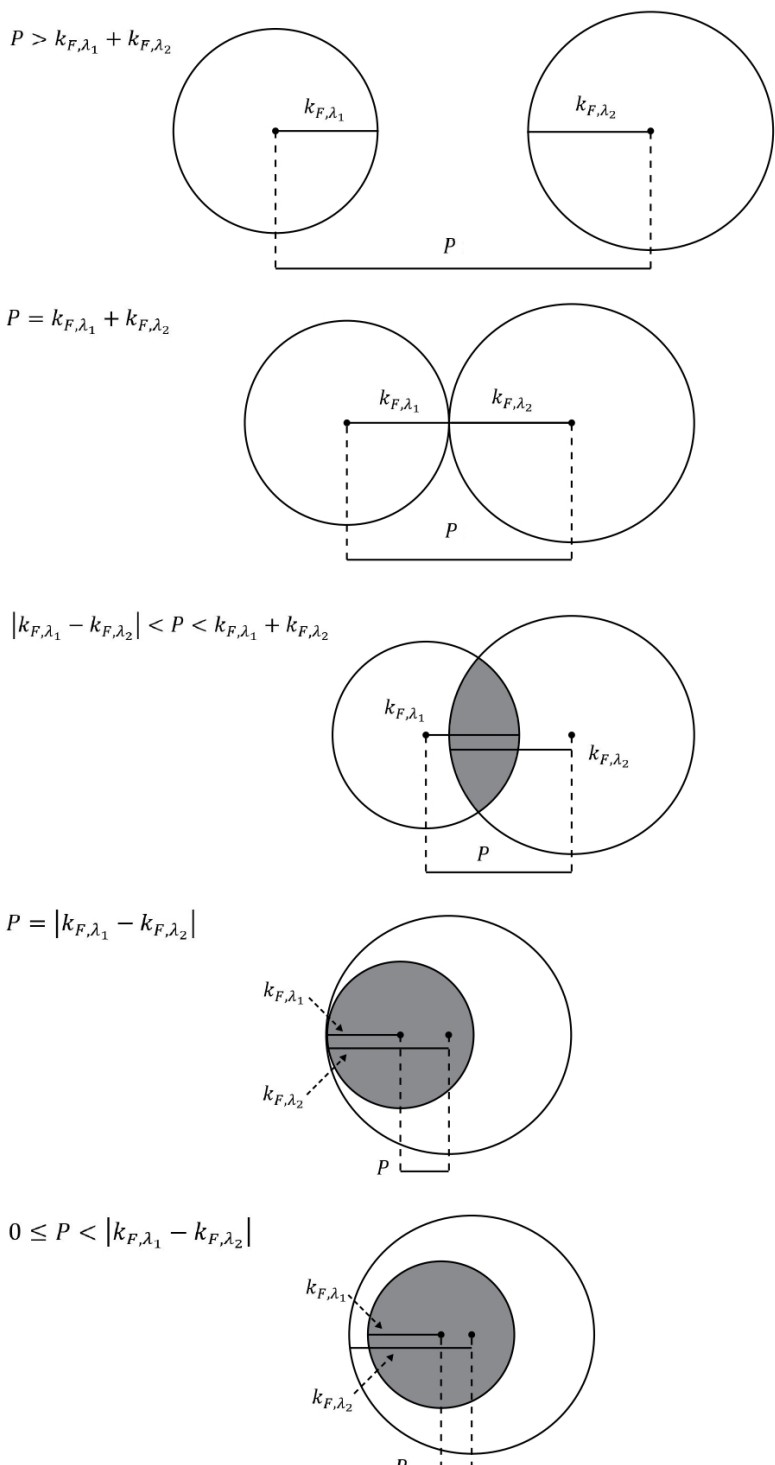

Figure 11: Sphere-sphere intersections. The grey area is the intersection we are looking for. $P$ is the distance between centers. $k_{F,\lambda_1}$ and $k_{F,\lambda_2}$ are the respective radii.

Where $Q_{x,y,z}$ are the coordinates in the initial configuration, and $Q_{\alpha x, \alpha y, \alpha z}$ are the new coordinates after the rotation.

Equations (A.9) and (A.10), which were the ones defining the two spheres, become in the new system of coordinates: (the sub-index $\alpha$ of $Q$ is omitted from now on)

$$I: \quad Q_x^2 + Q_y^2 + \left(Q_z + \frac{P}{2}\right)^2 \le k_{F,\lambda_1}^2, \tag{A.14}$$

$$II: \quad Q_x^2 + Q_y^2 + \left(Q_z - \frac{P}{2}\right)^2 \le k_{F,\lambda_2}^2. \tag{A.15}$$

Changing to cylindrical coordinates,

$$I: \quad r^2 + \left(z + \frac{P}{2}\right)^2 \le k_{F,\lambda_1}^2 \quad \Rightarrow \quad r_{max}[z] = \sqrt{k_{F,\lambda_1}^2 - \left(z + \frac{P}{2}\right)^2}, \tag{A.16}$$

$$II: \quad r^2 + \left(z - \frac{P}{2}\right)^2 \le k_{F,\lambda_2}^2 \quad \Rightarrow \quad r_{max}[z] = \sqrt{k_{F,\lambda_2}^2 - \left(z - \frac{P}{2}\right)^2}. \tag{A.17}$$

From the expressions above we can find the value of $z$ at which both spheres coincide, that is the value of $z$ that will separate the two spherical caps we need to integrate,

$$z_{\lim} = \frac{k_{F,\lambda_1}^2 - k_{F,\lambda_2}^2}{2P}. \tag{A.18}$$

We point out that $P$ can only be zero if both $k_{F,\lambda}$'s are equal, and in that case $z_{\lim}$ would exactly be 0.

After all these manipulations, now we are ready to write the final expression of the integral and the two regions of integration ( Spherical Caps Integral -SCI-)

$$SCI = \int r\,dr\,d\theta\,dz \frac{n_I n_{II}}{r^2 + z^2 - \frac{(p-p')^2}{4}}, \tag{A.19}$$

$$I: \quad r \in \left[0, \sqrt{k_{F,\lambda_1}^2 - \left(z + \frac{P}{2}\right)^2}\right], \qquad II: \quad r \in \left[\sqrt{k_{F,\lambda_2}^2 - \left(z - \frac{P}{2}\right)^2}\right],$$

$$\theta \in [0, 2\pi], \qquad\qquad \theta \in [0, 2\pi], \tag{A.20}$$

$$z \in \left[\frac{k_{F,\lambda_1}^2 - k_{F,\lambda_2}^2}{2P}, k_{F,\lambda_1} - \frac{P}{2}\right], \qquad z \in \left[\frac{P}{2} - k_{F,\lambda_2}, \frac{k_{F,\lambda_1}^2 - k_{F,\lambda_2}^2}{2P}\right].$$

The final step is to integrate and sum. After some algebra, the final integrated expression

is

$$SCI(P,Q,k_{F,\lambda_1},k_{F,\lambda_2}) = 2\pi\frac{1}{2}\left\{k_{F,\lambda_1} + k_{F,\lambda_2} - P + Q\ln\left|\frac{Q - k_{F,\lambda_1} + P/2}{Q + k_{F,\lambda_1} - P/2}\right|\right.$$
$$+ Q\ln\left|\frac{Q - k_{F,\lambda_2} + P/2}{Q + k_{F,\lambda_2} - P/2}\right|$$
$$+ \frac{(Q^2 + P^2/4 - k_{F,\lambda_1}^2)}{P}\ln\left|\frac{(k_{F,\lambda_1} - P/2)^2 - Q^2}{\frac{k_{F,\lambda_1}^2 + k_{F,\lambda_2}^2}{2} - \frac{P^2}{4} - Q^2}\right|$$
$$\left.+ \frac{(Q^2 + P^2/4 - k_{F,\lambda_2}^2)}{P}\ln\left|\frac{(k_{F,\lambda_2} - P/2)^2 - Q^2}{\frac{k_{F,\lambda_1}^2 + k_{F,\lambda_2}^2}{2} - \frac{P^2}{4} - Q^2}\right|\right\}. \tag{A.21}$$

Following the same reasoning we did in SI, at $P = 0$ the expression we have obtained can diverge. If $k_{F,\lambda_1}$ is equal to $k_{F,\lambda_2}$, then $P$ can be 0. In that limit, SCI becomes

$$SCI(0,Q,k_{F,\lambda_1} = k_{F,\lambda}, k_{F,\lambda_1} = k_{F,\lambda}) = 2\pi\left\{2k_{F,\lambda} + Q\ln\left|\frac{Q - k_{F,\lambda}}{Q + k_{F,\lambda}}\right|\right\}. \tag{A.22}$$

A summary of the different integration domains is shown in Table 2.

We have been able to solve analytically SI and SII, however, now we should square it and integrate it again. This second part has not been analytically achieved and numerical integration has led to the solution. This is written as

$$E_3(C_{\lambda_1}, C_{\lambda_2}) = \frac{1}{k_F^8}\int d\mathbf{p}n_p\int d\mathbf{p}'n_{p'}\left[SI(P,Q,k_{F,\lambda_1}) + SI(P,Q,k_{F,\lambda_2}) - SII(P,Q,k_{F,\lambda_1},k_{F,\lambda_2})\right]^2. \tag{A.23}$$

The final integrals can be simplified because the functions SI and SII only depend on the modules of $\mathbf{p}$ and $\mathbf{p}'$ and the angle $\theta$ between them. Moreover, if we perform the following change of variables $k_{F,\lambda_1} = k_F C_\lambda^{1/3}$, we can get rid of the Fermi momenta and express everything in terms of $C_\lambda$,

$$E_3(C_{\lambda_1}, C_{\lambda_2}) = 8\pi^2\int_0^{C_{\lambda_1}^{1/3}} p^2 dp \int_0^{C_{\lambda_2}^{1/3}} p'^2 dp' \int_{-1}^1 dx\left[SI(P,Q,C_{\lambda_1}^{1/3})\right.$$
$$\left.+ SI(P,Q,C_{\lambda_2}^{1/3}) - SII(P,Q,C_{\lambda_1}^{1/3},C_{\lambda_2}^{1/3})\right]^2, \tag{A.24}$$

with $x = -\cos\theta$, $P = \sqrt{p^2 + p'^2 - 2pp'x}$, and $2Q = \sqrt{p^2 + p'^2 + 2pp'x}$.

Table 2: Summary of the SII integral depending on the value of P.

| Condition | SII Integral |
|---|---|
| $0 \leq P \leq \|k_{F,\lambda_1} - k_{F,\lambda_2}\|$ | $SI(P,Q,min(k_{F,\lambda_1}, k_{F,\lambda_2}))$ |
| $\|k_{F,\lambda_1} - k_{F,\lambda_2}\| < P < (k_{F,\lambda_1} + k_{F,\lambda_2})$ | $SCI(P,Q,k_{F,\lambda_1}, k_{F,\lambda_2})$ |
| $P \geq (k_{F,\lambda_1} + k_{F,\lambda_2})$ | 0 |

### A.2   Term E4

The E4 contribution to the energy is

$$\frac{E}{N} = \epsilon_F \frac{3x^3}{32\pi^7} \frac{1}{\nu} \sum_{\lambda_1,\lambda_2} (1 - \delta_{\lambda_1,\lambda_2}) E_4(C_{\lambda_1}, C_{\lambda_2}), \tag{A.25}$$

with

$$E_4(C_{\lambda_1}, C_{\lambda_2}) = \frac{1}{k_F^8} \int d\mathbf{m}(1 - n_m) \int d\mathbf{m}'(1 - n_{m'}) \left[ \int 2\, d\mathbf{p} d\mathbf{p}' \frac{n_p n_{p'} \delta(\mathbf{p} + \mathbf{p}' - \mathbf{m} - \mathbf{m}')}{p^2 + p'^2 - m^2 - m^2} \right]^2. \tag{A.26}$$

The vectors $\mathbf{m}$ and $\mathbf{p}$ run over $k_{F,\lambda_1}$, and $\mathbf{m}'$ and $\mathbf{p}'$ run over $k_{F,\lambda_2}$. This term is similar to the previous term E3, but what is now different is the fact that the external integrals go to infinity and that the inner part is only the Sphere-Intersection-Integral (SII). Hence, we can rewrite it in terms of this known function (See Table 2),

$$E_4(C_{\lambda_1}, C_{\lambda_2}) = \frac{1}{k_F^8} \int d\mathbf{m}(1 - n_m) \int d\mathbf{m}'(1 - n_{m'}) \left[ SII(P, Q, k_{F,\lambda_1}, k_{F,\lambda_2}) \right]^2. \tag{A.27}$$

As we have done for E3, we can introduce $C_\lambda$ and simplify the external integrals,

$$E_4(C_{\lambda_1}, C_{\lambda_2}) = 8\pi^2 \int_0^{C_{\lambda_1}^{1/3} + C_{\lambda_2}^{1/3}} P^2 dP \int_{-\infty}^{\infty} dz \int_{r_{min}}^{\infty} r\, dr \left[ SII(P, Q, C_{\lambda_1}^{1/3}, C_{\lambda_2}^{1/3}) \right]^2, \tag{A.28}$$

with $\mathbf{P} = \mathbf{m} + \mathbf{m}'$ and $\mathbf{Q} = (\mathbf{m} - \mathbf{m}')/2$. Afterwards, the integrals in $\mathbf{Q}$ are transformed into cylindrical coordinates. The three angular integrals can be done trivially. The relation between $Q$, $r$ and $z$ is $Q = \sqrt{r^2 + z^2}$. The volume of integration of $\mathbf{Q}$ comes from $(1 - n_m)(1 - n_{m'}) = (1 - n_{P/2+Q})(1 - n_{P/2-Q})$, which is the space out of the intersection of two spheres. From here, we can know the integration limits of $r$ and $z$. The minimum value that $r$ can take depends on $P$ and $z$, as shown in Table 3.

### A.3   Term E5

The last term that contributes to the third-order expansion is a three-body interaction term,

$$\frac{E}{N} = \epsilon_F \frac{3x^3}{32\pi^7} \frac{1}{\nu} \sum_{\lambda_1,\lambda_2,\lambda_3} (1 - \delta_{\lambda_1,\lambda_2})(2 - 3\delta_{\lambda_1,\lambda_3} - 3\delta_{\lambda_2,\lambda_3}) E_5(C_{\lambda_1}, C_{\lambda_2}, C_{\lambda_3}), \tag{A.29}$$

with

$$\begin{aligned}
E_5(C_{\lambda_1}, C_{\lambda_2}, C_{\lambda_3}) = \frac{1}{2k_F^8} \Bigg\{ & \int d\mathbf{p}\, n_p \int d\mathbf{m}(1 - n_m) \left[ \int 2 d\mathbf{m}' d\mathbf{p}' \frac{(1 - n_{m'}) n_{p'}}{p^2 + p'^2 - m^2 - m'^2} \right. \\
& \times \delta(\mathbf{p} + \mathbf{p}' - \mathbf{m} - \mathbf{m}') \Bigg] \left[ \int 2 d\mathbf{m}_1 d\mathbf{p}_1 \frac{(1 - n_{m_1}) n_{p_1}}{p^2 + p_1^2 - m^2 - m_1^2} \delta(\mathbf{p} + \mathbf{p}_1 - \mathbf{m} - \mathbf{m}_1) \right] \\
& + \int d\mathbf{p}'\, n_{p'} \int d\mathbf{m}'(1 - n_{m'}) \left[ \int 2 d\mathbf{m} d\mathbf{p} \frac{(1 - n_m) n_p}{p^2 + p'^2 - m^2 - m'^2} \right. \\
& \times \delta(\mathbf{p} + \mathbf{p}' - \mathbf{m} - \mathbf{m}') \Bigg] \left[ \int 2 d\mathbf{m}_1 d\mathbf{p}_1 \frac{(1 - n_{m_1}) n_{p_1}}{p'^2 + p_1^2 - m'^2 - m_1^2} \right. \\
& \times \delta(\mathbf{p}' + \mathbf{p}_1 - \mathbf{m}' - \mathbf{m}_1) \Bigg] \Bigg\}.
\end{aligned} \tag{A.30}$$

Table 3: Values of $r_{min}$ depending on $P$ and $z$. The object $k_M$ is defined as $k_M = max(k_{F,\lambda_1}, k_{F,\lambda_2})$.

| Condition | $r_{min}$ |
|:---:|:---:|
| $P \le \|k_{F,\lambda_1} - k_{F,\lambda_2}\|$ & $z \in \left[ \dfrac{P}{2} - k_M, \dfrac{P}{2} + k_M \right]$ | $\sqrt{k_M^2 - \left(z - \dfrac{P}{2}\right)^2}$ |
| $\|k_{F,\lambda_1} - k_{F,\lambda_2}\| < P < (k_{F,\lambda_1} + k_{F,\lambda_2})$ & $z \in \left[ -\dfrac{P}{2} - k_{F,\lambda_1}, \dfrac{k_{F,\lambda_1}^2 - k_{F,\lambda_2}^2}{2P} \right]$ | $\sqrt{k_{F,\lambda_1}^2 - \left(z + \dfrac{P}{2}\right)^2}$ |
| $\|k_{F,\lambda_1} - k_{F,\lambda_2}\| < P < (k_{F,\lambda_1} + k_{F,\lambda_2})$ & $z \in \left[ \dfrac{k_{F,\lambda_1}^2 - k_{F,\lambda_2}^2}{2P}, \dfrac{P}{2} + k_{F,\lambda_2} \right]$ | $\sqrt{k_{F,\lambda_2}^2 - \left(z - \dfrac{P}{2}\right)^2}$ |
| Else | $0$ |

One identifies four inner integrals that have the same formal shape. Each one is of the form

$$I_{5,\text{inner}} = \left[ \int 2d\mathbf{m}'d\mathbf{p}' \frac{(1-n_{m'})n_{p'}}{p^2 + p'^2 - m^2 - m'^2} \delta(\mathbf{p} + \mathbf{p}' - \mathbf{m} - \mathbf{m}') \right]. \tag{A.31}$$

In this integral, we decompose the term $(1 - n_{m'})n_{p'} = n_{p'} - n_{m'}n_{p'}$, so we have again a sphere and the intersection of two spheres. One can proceed then in a similar way to previous integrals. The spherical part ($SI_5$) is

$$\begin{aligned}
SI_5 &= \int 2d\mathbf{m}'d\mathbf{p}' \frac{n_{p'}}{p^2 + p'^2 - m^2 - m'^2} \delta(\mathbf{p} + \mathbf{p}' - \mathbf{m} - \mathbf{m}') \\
&= -\int d\mathbf{p}' \frac{n_{p'}}{m^2 + \mathbf{p}' \cdot (\mathbf{p} - \mathbf{m}) - \mathbf{p} \cdot \mathbf{m}} \\
&= 2\pi \int_0^{k_{F,\lambda}} p'^2 dp' \frac{1}{p'|\mathbf{m} - \mathbf{p}|} \ln \left| \frac{m^2 - \mathbf{p} \cdot \mathbf{m} - p'|\mathbf{m} - \mathbf{p}|}{m^2 - \mathbf{p} \cdot \mathbf{m} + p'|\mathbf{m} - \mathbf{p}|} \right| \\
&= 2\pi \left\{ -k_{F,\lambda} \frac{(m^2 - \mathbf{p} \cdot \mathbf{m})}{|\mathbf{m} - \mathbf{p}|^2} + \left( \frac{k_{F,\lambda}^2}{2|\mathbf{m} - \mathbf{p}|} - \frac{(m^2 - \mathbf{p} \cdot \mathbf{m})^2}{2|\mathbf{m} - \mathbf{p}|^3} \right) \ln \left| \frac{m^2 - \mathbf{p} \cdot \mathbf{m} - k_{F,\lambda}|\mathbf{m} - \mathbf{p}|}{m^2 - \mathbf{p} \cdot \mathbf{m} + k_{F,\lambda}|\mathbf{m} - \mathbf{p}|} \right| \right\}.
\end{aligned} \tag{A.32}$$

It can be rewritten as

$$SI_5 = 2\pi \left\{ -k_{F,\lambda} \frac{(2Q^2 - a)}{4Q^2} + \left( \frac{k_{F,\lambda}^2}{4Q} - \frac{(2Q^2 - a)^2}{16Q^3} \right) \ln \left| \frac{2Q^2 - a - 2k_{F,\lambda}Q}{2Q^2 - a + 2k_{F,\lambda}Q} \right| \right\}, \tag{A.33}$$

with $a = (p^2 - m^2)/2$ and $Q = |\mathbf{m} - \mathbf{p}|/2$.

Before doing the sphere intersection part, we define a change of variables that are useful

for this particular integral,

$$\mathbf{s} = \frac{\mathbf{p}' + \mathbf{m}'}{2}, \qquad \mathbf{p}' = \mathbf{s} + \frac{\mathbf{d}}{2},$$
$$\Rightarrow$$
$$\mathbf{d} = \mathbf{p}' - \mathbf{m}' \qquad \mathbf{m}' = \mathbf{s} - \frac{\mathbf{d}}{2}. \tag{A.34}$$

Then,

$$
\begin{aligned}
SII_5 &= \int 2 d\mathbf{m}' d\mathbf{p}' \frac{n_{m'} n_{p'}}{p^2 + p'^2 - m^2 - m'^2} \delta(\mathbf{p} + \mathbf{p}' - \mathbf{m} - \mathbf{m}') \\
&= \int ds d\mathbf{d} \frac{n_{s+d/2} n_{s-d/2}}{\dfrac{p^2 - m^2}{2} + \mathbf{s} \cdot \mathbf{d}} \delta(\mathbf{p} - \mathbf{m} + \mathbf{d}) \\
&= \int ds \frac{n_{s+d/2} n_{s-d/2}}{\dfrac{p^2 - m^2}{2} + \mathbf{s} \cdot \mathbf{d}}, \quad \text{where} \quad \mathbf{d} = \mathbf{m} - \mathbf{p}.
\end{aligned}
\tag{A.35}
$$

As commented previously, there are three types of intersection: null, two caps and a small sphere. The null is just zero and the small-sphere case does not happen here because the two radius are equal. Therefore just one case is missing, the one concerning the two spherical caps. So $SII_5$ will be just $SCI_5$. After some algebra, one arrives to

$$
\begin{aligned}
SCI_5 = 2\pi \Bigg\{ &\frac{8ak_{F,\lambda}d - 4ad^2}{8d^3} - \frac{(4a^2 - 4ad^2 - 4k_{F,\lambda}^2 d^2 + d^4)}{8d^3} \ln \left| \frac{a + (k_{F,\lambda} - d/2)d}{a} \right| \\
&+ \frac{(4a^2 + 4ad^2 - 4k_{F,\lambda}^2 d^2 + d^4)}{8d^3} \ln \left| \frac{a + (d/2 - k_{F,\lambda})d}{a} \right| \Bigg\}.
\end{aligned}
\tag{A.36}
$$

Changing variable to $Q = |\mathbf{m} - \mathbf{p}|/2 = d/2$, we get

$$
\begin{aligned}
SCI_5 = 2\pi \Bigg\{ &\frac{ak_{F,\lambda}Q - aQ^2}{4Q^3} - \frac{(a^2 - 4aQ^2 - 4k_{F,\lambda}^2 Q^2 + 4Q^4)}{16Q^3} \ln \left| \frac{a + 2(k_{F,\lambda} - Q)Q}{a} \right| \\
&+ \frac{(a^2 + 4aQ^2 - 4k_{F,\lambda}^2 Q^2 + 4Q^4)}{16Q^3} \ln \left| \frac{a + 2(Q - k_{F,\lambda})Q}{a} \right| \Bigg\},
\end{aligned}
\tag{A.37}
$$

with $a = (p^2 - m^2)/2$.

Therefore, $I_{5,\text{inner}}$ is $SI_5 - SCI_5$, but in both functions there is one situation that can cause problems, and that is the limit when $Q \to 0$. However, if one calculates separately the limits for $SI_5$ and $SCI_5$, both give $4\pi k_{F,\lambda}^3/(3a)$, so the total limit is zero. The possible contributions to $I_{5,\text{inner}}$ are summarized in Table 4.

If we look to $SI_5$ and $SCI_5$, we can see that the integrals depend on $k_{F,\lambda}$, $a$ and $Q$. Hence, they depend on the modulus of both $\mathbf{m}$ and $\mathbf{p}$, and also on the relative angle between both vectors. We recall that $a = (p^2 - m^2)/2$ and $Q = \sqrt{p^2 + m^2 + 2pmx}/2$ with $x = -\cos\theta$. In Eq. (A.38), we show the entire expression of $E_5$, the inner integrals are expressed through the

Table 4: Summary of the $SI_{5,\text{inner}}$ integral depending on the value of $Q$.

| Condition | $I_{5,\text{inner}}$ **Integral** |
|:---:|:---:|
| $Q = 0$ | $0$ |
| $0 < Q < k_{F,\lambda}$ | $SI_5 - SCI_5$ |
| $Q \geq k_{F,\lambda}$ | $SI_5$ |

compact form $I_{5,inner}(R, a, k_{F,\lambda})$.

$$E_5(C_{\lambda_1}, C_{\lambda_2}, C_{\lambda_3}) = \frac{1}{2k_F^8}\left\{\int d\mathbf{p}\, n_p \int d\mathbf{m}(1-n_m)I_{5,inner}(R,a,k_{F,\lambda_2})I_{5,inner}(R,a,k_{F,\lambda_3})\right.$$
$$\left. + \int d\mathbf{p}'\, n_{p'} \int d\mathbf{m}'(1-n_{m'})I_{5,inner}(R',a',k_{F,\lambda_1})I_{5,inner}(R',a',k_{F,\lambda_3})\right\}. \tag{A.38}$$

We point out that $R'$ means $\sqrt{p'^2 + m'^2 + 2p'm'x'}/2$ and $a' = (p'^2 - m'^2)/2$. In terms of the concentrations $C_\lambda$ one arrives to the final expression

$$E_5(C_{\lambda_1}, C_{\lambda_2}, C_{\lambda_3}) = 4\pi^2\left\{\int_0^{C_{\lambda_1}^{1/3}} dp \int_{C_{\lambda_1}^{1/3}}^\infty dm \int_{-1}^1 dx\, I_{5,\text{inner}}(Q,a,k_{F,\lambda_2})I_{5,\text{inner}}(Q,a,k_{F,\lambda_3})\right.$$
$$\left. + \int_0^{C_{\lambda_2}^{1/3}} dp' \int_{C_{\lambda_2}^{1/3}}^\infty dm' \int_{-1}^1 dx'\, I_{5,\text{inner}}(Q',a',k_{F,\lambda_1})I_{5,\text{inner}}(Q',a',k_{F,\lambda_3})\right\}. \tag{A.39}$$

We would like to point out that for the three integrals ($E3$, $E4$, and $E5$) we have been able to reduce high dimensional integrals to just 3D integrals, which makes numerical integration less costly.

# B  Terms E3, E4, and E5 for the ground state

In Ref. [20], we proved that a Fermi gas in the ground state chooses a certain spin occupational distribution. This configuration, that minimizes the energy of the system, is the following: one species increases, and the rest ones diminish equally. Under these conditions, the concentrations $C_\lambda$ for a given polarization $P$ are

$$C_+ = 1 + |P|(\nu - 1), \tag{B.1}$$
$$C_- = 1 - |P|, \tag{B.2}$$

with subindex $+$ standing for the state with the larger population and $-$ for the rest. We point out that, in this configuration, all the species that decrease have the same Fermi momentum or, equivalently, the same concentration $C_\lambda$. This is important because it reduces the number of integrals to be carried out. As commented before, the energy corresponding to these terms is

$$\frac{E}{N} = \frac{3}{5}\epsilon_F\left[\frac{1}{\nu}\sum_{\lambda_1,\lambda_2}\left\{\frac{5}{32\pi^7}\left(E_3 + E_4 + \sum_{\lambda_3}(2 - 3\delta_{\lambda_1,\lambda_3} - 3\delta_{\lambda_2,\lambda_3})E_5\right)(k_F a_0)^3\right\}(1-\delta_{\lambda_1,\lambda_2})\right]. \tag{B.3}$$

For $E3$ and $E4$, we only have to do a sum over pairs. Therefore, we only have two terms, one taking into account the combination $+-$, and the other one accounting for $--$,

$$\frac{1}{2}\sum_{\lambda_1,\lambda_2} E_3(C_{\lambda_1}, C_{\lambda_2})(1-\delta_{\lambda_1,\lambda_2}) = (\nu-1)E_3(C_+, C_-) + \frac{(\nu-1)(\nu-2)}{2}E_3(C_-, C_-). \quad \text{(B.4)}$$

Moreover, the second term can be simplified because both concentrations are the same. The function $E_3(C_-, C_-)$ is the same as its value at zero polarization times $C_-^{8/3}$,

$$\frac{1}{2}\sum_{\lambda_1,\lambda_2} E_3(C_{\lambda_1}, C_{\lambda_2})(1-\delta_{\lambda_1,\lambda_2}) = (\nu-1)E_3(C_+, C_-) + \frac{(\nu-1)(\nu-2)}{2}E_3(1, 1)C_-^{8/3}. \quad \text{(B.5)}$$

The same reasoning done for $E3$ can be applied to $E4$,

$$\frac{1}{2}\sum_{\lambda_1,\lambda_2} E_4(C_{\lambda_1}, C_{\lambda_2})(1-\delta_{\lambda_1,\lambda_2}) = (\nu-1)E_4(C_+, C_-) + \frac{(\nu-1)(\nu-2)}{2}E_4(1, 1)C_-^{8/3}. \quad \text{(B.6)}$$

For $E5$, as we have a three-body interaction, there are more possible combinations, but they can be grouped up to four terms,

$$\frac{1}{2}\sum_{\lambda_1,\lambda_2,\lambda_3} E_5(C_{\lambda_1}, C_{\lambda_2}, C_{\lambda_3})(1-\delta_{\lambda_1,\lambda_2})(2-3\delta_{\lambda_1,\lambda_3}-3\delta_{\lambda_2,\lambda_3}) = -(\nu-1)E_5(C_+, C_-, C_+)$$
$$+ (\nu-1)(2\nu-5)E_5(C_+, C_-, C_-) + (\nu-1)(\nu-2)E_5(C_-, C_-, C_+)$$
$$+ (\nu-1)(\nu-2)(\nu-4)E_5(C_-, C_-, C_-). \quad \text{(B.7)}$$

In the same manner as E3 and E4, the term $E_5(C_-, C_-, C_-)$ can be simplified into $E_5(1,1,1)C_-^{8/3}$, which is the value at zero polarization times $C_-^{8/3}$,

$$\frac{1}{2}\sum_{\lambda_1,\lambda_2,\lambda_3} E_5(C_{\lambda_1}, C_{\lambda_2}, C_{\lambda_3})(1-\delta_{\lambda_1,\lambda_2})(2-3\delta_{\lambda_1,\lambda_3}-3\delta_{\lambda_2,\lambda_3}) = -(\nu-1)E_5(C_+, C_-, C_+)$$
$$+ (\nu-1)(2\nu-5)E_5(C_+, C_-, C_-) + (\nu-1)(\nu-2)E_5(C_-, C_-, C_+)$$
$$+ (\nu-1)(\nu-2)(\nu-4)E_5(1, 1, 1)C_-^{8/3}. \quad \text{(B.8)}$$

Therefore, in the ground state, of all the integrals we had to calculate, we only care about five, which are: $E_3(C_+, C_-)$, $E_4(C_+, C_-)$, $E_5(C_+, C_-, C_+)$, $E_5(C_+, C_-, C_-)$, and $E_5(C_-, C_-, C_+)$.

## C  Functions E3, E4, and E5 for different spins

In this section, we plot the integrals obtained numerically for different spin values. To simplify the notation, we use E3 for $E_3(C_+, C_-)$, E4 for $E_4(C_+, C_-)$, E5_1 for $E_5(C_+, C_-, C_+)$, E5_2 for $E_5(C_+, C_-, C_-)$, and E5_3 for $E_5(C_-, C_-, C_+)$.

## C.1 Spin 1/2

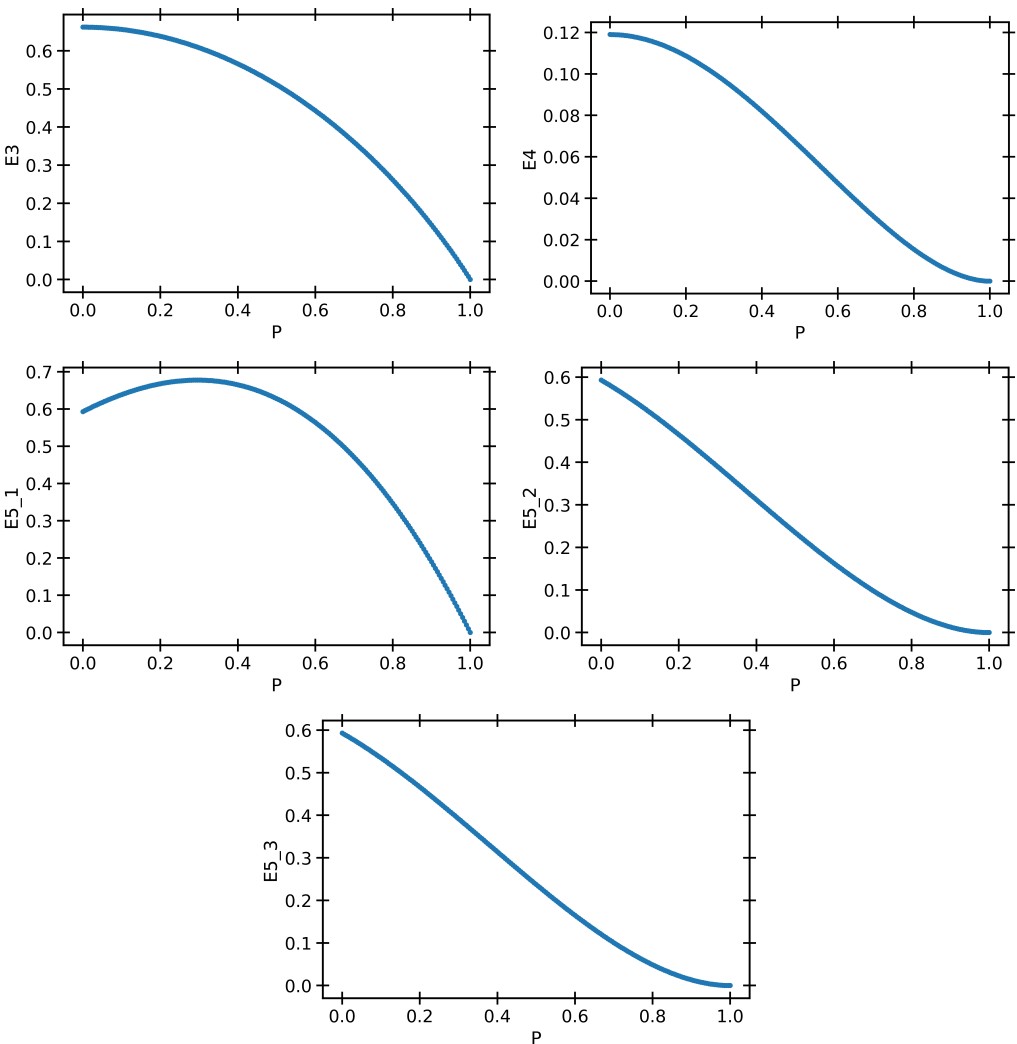

Figure 12: E3, E4, E5_1, E5_2 and E5_3 in terms of the polarization $P$ for $S = 1/2$. The error bars are smaller than the size of the symbols.

## C.2 Spin 3/2

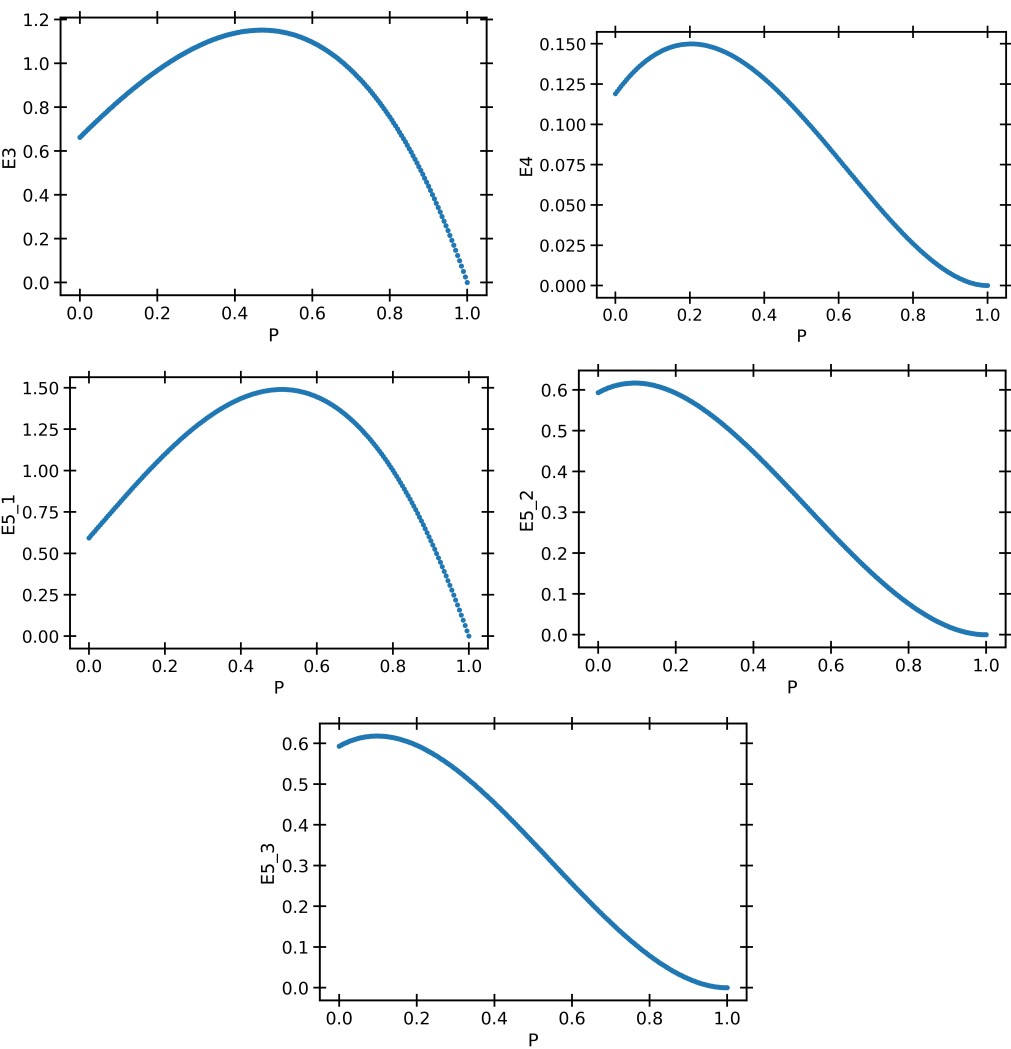

Figure 13: E3, E4, E5_1, E5_2 and E5_3 in terms of the polarization $P$ for $S = 3/2$. The error bars are smaller than the size of the symbols.

## C.3 Spin 5/2

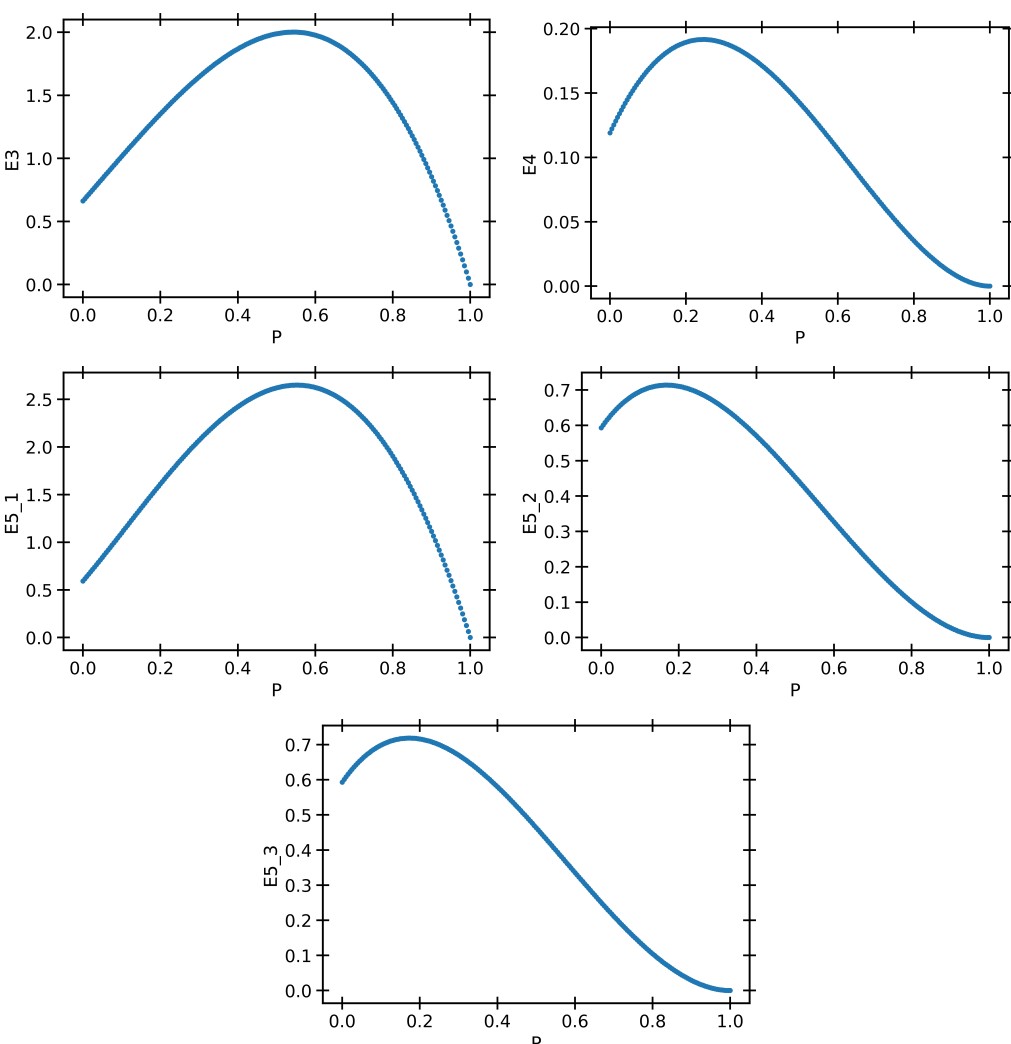

Figure 14: E3, E4, E5_1, E5_2 and E5_3 in terms of the polarization $P$ for $S = 5/2$. The error bars are smaller than the size of the symbols.

## C.4 Spin 7/2

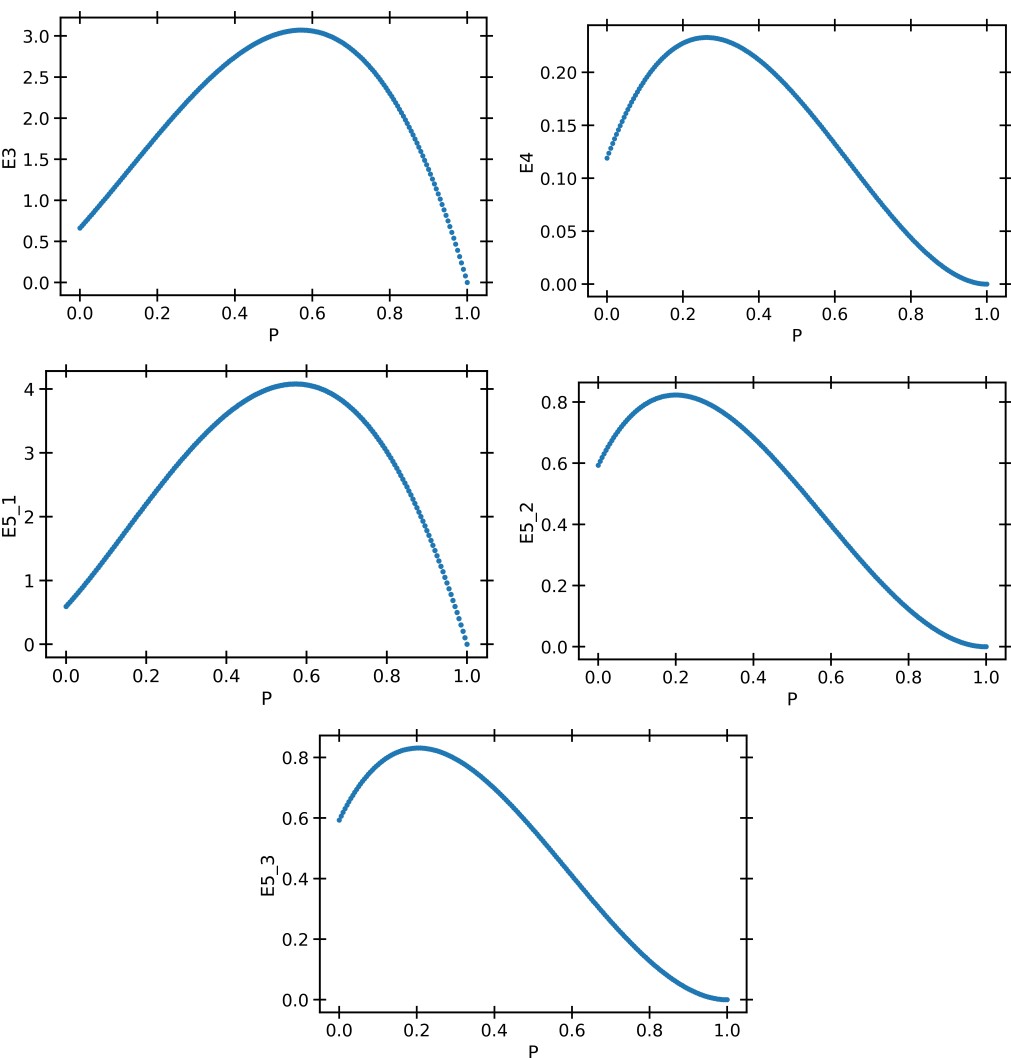

Figure 15: E3, E4, E5_1, E5_2 and E5_3 in terms of the polarization $P$ for $S = 7/2$. The error bars are smaller than the size of the symbols.

## C.5 Spin 9/2

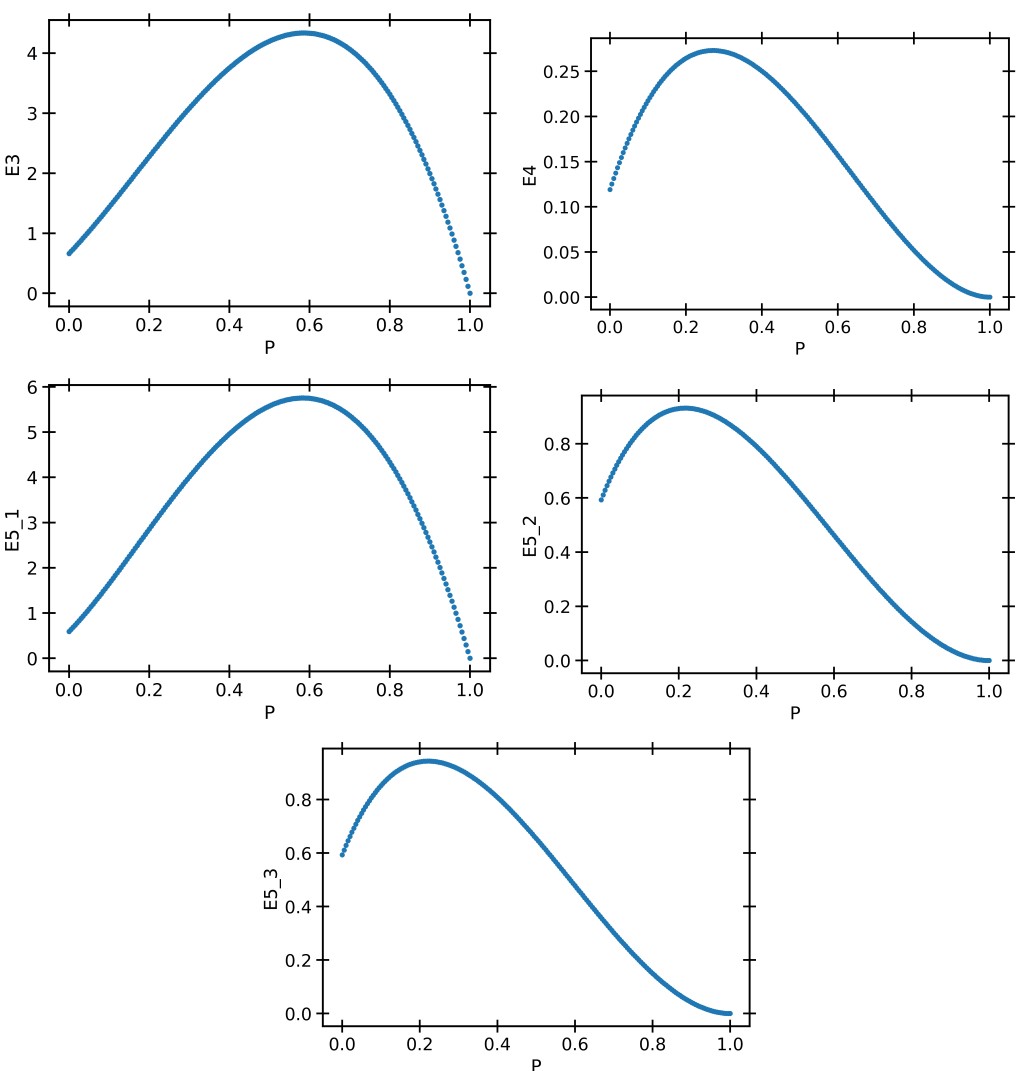

Figure 16: E3, E4, E5_1, E5_2 and E5_3 in terms of the polarization $P$ for $S = 9/2$. The error bars are smaller than the size of the symbols.

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
