# Peer review of "Beyond universality in repulsive SU(N) Fermi gases"

_SciPost Physics, doi:SciPost Phys. 17, 030 (2024)_

## Round 3 · Referee Report · Anonymous (Referee 2) · 2024-1-24

Report

The authors have exhaustively addressed the comments I raised in my previous report. In particular, they clarified the definition of quasi-continuous transition, they provided additional information on the adaptive Monte Carlo integration, and, interestingly, they further discussed the effects of intra-species interactions. I recommend publication of the manuscript in SciPost Physics.

  • validity: high
  • significance: high
  • originality: good
  • clarity: -
  • formatting: excellent
  • grammar: excellent

Author:  Jordi Boronat  on 2024-01-24  [id 4276]

(in reply to Report 1 on 2024-01-24)

We acknowledge the positive comments of the Referee on our work and his/her recommendation for acceptance in Scipost Physics. We hope that now the paper is ready for acceptance.

---

## Round 3 · Referee Report · Anonymous (Referee 3) · 2024-5-15

Strengths

1- The article provides a comprehensive perturbative description of the energy of a fermionic gas of interacting particles with spin $S=N/2$. Each term is thoroughly described, and the analytical description is general enough to be applicable to softcore and hardcore particles.
2- The results of the article are carefully benchmarked to numerical results of Monte-Carlo simulations, displaying a very sound agreement. Furthermore, discrepancies between results are also discussed.
4- The predictions formulated by the authors on the critical behavior of the ferromagnetic transition in $SU(N)$ Fermi gases can be readily investigated in existing experimental set-ups.

Weaknesses

1- The article is ambiguous regarding the numerical method, or the parameter regimes used to extract some results.
2- The type of transitions and the phases that can be encountered as a function of $S$ are treated in a rather superficial manner.

Report

In this article, the authors investigate itinerant ferromagnetism in a dilute Fermi gas. To this end, they derive an analytical expression for the energy per particle via a perturbative diagrammatic expansion up to the third order in the parameters of the problem, namely the $s-$ and $p-$wave scattering lengths and the $s-$wave effective range. In particular, the spin degrees of freedom are considered for an arbitrary spin length. Then, based on the derived expression, the authors determine the magnetization of the ground state and observe as a function of the gas parameter $x$ different types of transitions towards a ferromagnetic state, depending on the length $S$ of the spin. Finally, the authors characterize the emergence of itinerant ferromagnetism via the magnetization, the magnetic susceptibility and Tan's constant.

During the last decade or so, efforts made in experimental and theoretical physics converged in order to shed light on the physics of $SU(N)$ system, and this work is in line with this joint effort. In my opinion, its relevance for the description of available experiments on Yb and Sr gases as well as its good degree of agreement with Monte-Carlo simulation make this article suitable for publication in SciPost Physics.

However, before fully agreeing to a publication, I would like the authors to address two points:

-Firstly, the authors do not specify how they determine the magnetization displayed on Fig. 5. If it is evaluated by using the value of the order parameter $P$ that minimizes the energy $E$ for a given value of $x$, as shown on Fig. 3, then it should be explicitly stated in the manuscript. Similarly, the authors do not provide the values chosen for the parameters $a_1$ and $r_0$ on the softcore results shown on Fig. 10.

-Secondly, based on the behavior of the magnetization on Fig 5 and the susceptibility on Fig 7, it seems that for $S=7/2$ and $S=9/2$ the Fermi gas undergoes two phase transitions, namely a second-order phase transition from an unpolarized system and then a first-order phase transition towards a polarized phase. This interpretation of the results, that is also further supported by the two singularities observed in the magnetic susceptibility, hints at the existence of an intermediate phase with some sort of magnetic ordering. Indeed, several examples can be found in the literature of $SU(N)$ systems featuring rich phase diagrams, such as in Nucl. Phys. B 996 (2023) 116353. Could the authors comment on this remark?

Requested changes

I invite the authors to correct the following typos
1- Top of right column of p2 : "require of a combination" -> "require a combination".
2- Left column on p3: "$s=1/2$" -> "$S=1/2$".
3- Right column on p6: "$k_F a_0=0,85$"-> "$k_F a_0=0.85$" and "$k_F a_0=0,9$"-> "$k_F a_0=0.9$".

4-On p6, the authors claim that the use plane-waves Slater determinant in Diffusion Monte Carlo describe more efficiently polarized Fermi gases. I would ask the authors to add a reference to support this statement.
5- On top of the right column of p7, could the authors clarify what they mean by "partial discontinuous transitions"?

Recommendation

Ask for minor revision

---

## Round 4 · Author Response

Response to Referee 2

We kindly thank the Referee for his/her positive comments and for finding our manuscript appropriate for SciPost Phys. In the following, we address the comments and suggestions pointed out by the Referee and the changes introduced in the
manuscript accordingly.

Concern 1:

Firstly, the authors do not specify how they determine the magnetization displayed on Fig. 5. If it is evaluated by using the value of the order parameter P that minimizes the energy E for a given value of $x$, as shown on Fig. 3, then it should be explicitly stated in the manuscript. Similarly, the authors do not provide the values chosen for the parameters $a_1$ and $r_0$ on the softcore results shown on Fig. 10.

Our response: The Referee is right. The polarization comes from the minimization of the energy. We have added a comment on that.

Concerning the second point, the values of $r_0$ and $a_1$ used in Fig. 10 are the same as the ones used in Fig. 2. We have mentioned that explicitly in the caption of Fig. 10 and also in the discussion of the results. Anyway, we have introduced a new sentence in the text to avoid any misunderstanding.

Concern 2:

Secondly, based on the behavior of the magnetization on Fig 5 and the susceptibility on Fig 7, it seems that for S=7/2 and S=9/2 the Fermi gas undergoes two phase transitions, namely a second-order phase transition from an unpolarized system and then a first-order phase transition towards a polarized phase. This interpretation of the results, that is also further supported by the two singularities observed in the magnetic susceptibility, hints at the existence of an intermediate phase with some sort of magnetic ordering. Indeed, several examples can be found in the literature of SU(N) systems featuring rich phase diagrams, such as in Nucl. Phys. B 996 (2023) 116353. Could the authors comment on this remark?

Our response: We thank the Referee for this observation. We have thus included a new comment about what happens between the peaks in the magnetic susceptibility.

Concern 3:

Top of right column of p2: "require of a combination" -> "require a combination".

Our response: Done.

Concern 4:

Left column on p3: "s=1/2" -> "S=1/2".

Our response: Done.

Concern 5:

Right column on p6: "$k_Fa_0=0,85$"-> "$k_Fa_0=0.85$" and "$k_Fa_0=0,9$"-> "$k_Fa_0=0.9$".

Our response: Done.

Concern 6:

On p6, the authors claim that the use plane-waves Slater determinant in Diffusion Monte Carlo describe more efficiently polarized Fermi gases. I would ask the authors to add a reference to support this statement.

Our response: Diffusion Monte Carlo works in the so called Fixed Node approximation where the nodes are fixed and correspond to the ones of the variational wave function used for importance sampling. Pandharipande and others proved that the first correction to the nodes, coming from a plane-wave Slater determinant, are the backflow correlations that change the position of the nodes due to the interparticle interactions. Pandharipande proved that backflow correlations are more relevant in normal liquid $^3$He than in polarized $^3$He because in the polarized phase s-wave collisions are forbidden. This is the reason behind the statement that the plane-wave model is worse in normal than in fully polarized Fermi liquids. In the new version of the manuscript, we have added two references on this particular issue.

Concern 7:

On top of the right column of p7, could the authors clarify what they mean by "partial discontinuous transitions"?

Our response: We have added a comment on that.

---

## Round 4 · List of Changes

1) In response to Concern 1 we have introduced the following sentences:

In order to gain a deeper understanding of the phase transition, we plot in Fig. 5 the order parameter (in
this instance, the polarization) as a function of the gas parameter $x$. The polarization we plot is the one that minimizes the energy at a given $x$.

We report the critical gas parameters obtained using both the hard-sphere and soft-sphere
potentials. The soft-sphere potential is the same we used in Fig. 2, this means $r_0=0.424\,a_0$ and $a_1=1.1333\,a_0$.

2) In response to Concern 2 we have introduced the following sentences:

For spins 7/2 and 9/2, we see the singular double transition (two peaks) that we have mentioned before. With increasing density, the first peak corresponds to the truncated continuous transition, and the second peak to the latter first-order phase transition. The two peaks point to the existence of an intermediate phase between the non-polarized phase and the fully-polarized one. This phase would be located around the bottom that lies between peaks in Fig.~7. And, according to Fig.~5, this intermediate phase would have a partial polarization, hence, the Fermi gas would exhibit some kind of magnetic ordering. The rich phase diagram that appears in SU(N) Fermi systems have also been pointed out in Ref.~[45].

3) In response to Concern 7 we have introduced the following sentence:

For spin $3/2$ and $5/2$, we have partial discontinuous transitions, as there is a polarization jump, but it does not reach $P=1$, hence, the label 'partial'. If it reached $P=1$ directly, it would be a total discontinuous transition.

---

## Editorial Decision

published